# Spatial transcriptomics reveals discrete tumour microenvironments and autocrine loops within ovarian cancer subclones

Elena Denisenko [1] ✉, Leanne de Kock [1,12], Adeline Tan[2], Aaron B. Beasley [3], Maria Beilin[4], Matthew E. Jones[1], Rui Hou[1], Dáithí Ó Muirí[1], Sanela Bilic[4], G. Raj K. A. Mohan[4,5], Stuart Salfinger[6], Simon Fox [1], Khaing P. W. Hmon[1], Yen Yeow [1], Youngmi Kim[7], Rhea John[7], Tami S. Gilderman[7], Emily Killingbeck [7], Elin S. Gray [3], Paul A. Cohen [8,9] ✉, Yu Yu[8,10,11] ✉ & Alistair R. R. Forrest [1] ✉

High-grade serous ovarian carcinoma (HGSOC) is genetically unstable and characterised by the presence of subclones with distinct genotypes. Intratumoural heterogeneity is linked to recurrence, chemotherapy resistance, and poor prognosis. Here, we use spatial transcriptomics to identify HGSOC subclones and study their association with infiltrating cell populations. Visium spatial transcriptomics reveals multiple tumour subclones with different copy number alterations present within individual tumour sections. These subclones differentially express various ligands and receptors and are predicted to differentially associate with different stromal and immune cell populations. In one sample, CosMx single molecule imaging reveals subclones differentially associating with immune cell populations, fibroblasts, and endothelial cells. Cell-to-cell communication analysis identifies subclone-specific signalling to stromal and immune cells and multiple subclone-specific autocrine loops. Our study highlights the high degree of subclonal heterogeneity in HGSOC and suggests that subclone-specific ligand and receptor expression patterns likely modulate how HGSOC cells interact with their local microenvironment.

Ovarian cancer is the eighth leading cause of cancer deaths in women worldwide[1]. High-grade serous ovarian carcinoma (HGSOC) is the most common and lethal histologic subtype, accounting for 70-80% of ovarian cancer deaths[2]. HGSOC is thought to be derived from both fallopian tube and ovarian surface epithelium[3,4] and genomically is characterised by almost universal *TP53* mutations and copy number alterations (CNAs)[5-8]. Notably, although several chromosomal regions are recurrently altered[5], and multiple genes (*FAT3, CSMD3, BRCA1,*

[1]Harry Perkins Institute of Medical Research, QEII Medical Centre and Centre for Medical Research, The University of Western Australia, Nedlands, Perth, WA 6009, Australia. [2]Anatomical Pathology Department, Clinipath, Sonic Healthcare, Perth, WA 6017, Australia. [3]Centre for Precision Health, Edith Cowan University, Joondalup, WA 6027, Australia. [4]Department of Gynaecological Oncology, Bendat Family Comprehensive Cancer Centre, St John of God Subiaco Hospital, 12 Salvado Rd, Subiaco, WA 6008, Australia. [5]School of Medicine, University of Notre Dame, Fremantle, WA 6160, Australia. [6]Western Australian Gynae and Surgery, Perth, WA, Australia. [7]NanoString Technologies, Seattle, WA, USA. [8]Division of Obstetrics and Gynaecology, Medical School, University of Western Australia, 35 Stirling Highway, Crawley, WA 6009, Australia. [9]Institute for Health Research, The University of Notre Dame Australia, 32 Mouat Street Fremantle, WA 6160, Australia. [10]Curtin Medical School, Curtin University, 410 Koorliny Way, Bentley, WA 6102, Australia. [11]Curtin Health Innovation Research Institute, Curtin University B305, Bentley, WA 6102, Australia. [12]Present address: Children's Hospital of Eastern Ontario Research Institute, Ottawa, ON, Canada. ✉e-mail: elena.denisenko@perkins.org.au; paul.cohen@uwa.edu.au; yu.yu@curtin.edu.au; alistair.forrest@gmail.com

*BRCA2, NF1, CDK12, GABRA6, RB1, PTEN*, and *RAD51B*) are recurrently disrupted, HGSOC genomes are highly heterogeneous with most of the above alterations only found in a small fraction of tumours[5,9–12]. Also, due to a high degree of chromosomal instability[13], most HGSOCs are polyclonal[14,15]. As the cancer progresses and metastasises, clonal diversity increases, which is associated with worse prognosis and development of chemoresistance[8,9,13,16,17].

Recently, a number of investigations employing single-cell RNA-sequencing (scRNA-seq) have explored the composition of cell types within the tumour microenvironment of HGSOC, both in primary tumours and metastatic sites[18–23]. Additionally, these studies utilised copy number inference techniques to detect chromosomal copy number alterations (CNAs) in HGSOC tumour cells[19,21] and subclones[24] exhibiting distinct CNAs. With these single cell profiles it is now apparent that previously reported transcriptional subtypes of HGSOC based on bulk expression measurements (mesenchymal, immunoreactive, differentiated, and proliferative), which are associated with differences in prognosis[25], largely reflect the degree of immune cell infiltration and the abundance of fibroblasts[19], rather than inherent differences in tumour cells. To determine how these non-malignant cell types might influence tumour growth and prognosis, several

groups have predicted ligand-receptor interactions between stromal, immune and tumour cell populations[23,24].

Here we use spatial transcriptomics (10x Genomics Visium and NanoString® CosMx™ Spatial Molecular Imaging (SMI)) of HGSOC tumours to reveal the relationship between tumour subclonal genotypes and infiltration patterns by non-malignant cell types. Using CNA inference, we predict that even within small tumour sections (<6.5mm²) several regionally distinct subclones can exist. We show that the subclones identified display different patterns of infiltration and that tumour cells may influence their local microenvironment by subclone-specific upregulation of ligands and receptors. We also find evidence of subclone-specific autocrine loops where ligands, cognate receptors, or both are upregulated in one clone compared to another. Our analyses predict a link between subclonal genotype differences and differential infiltration patterns.

## Results

### Spatial gene expression of HGSOC tumours
Visium spatial transcriptomics technology utilises a grid of ~5000 55 μm spots with uniquely barcoded oligo-dT primers, spaced 100 μm apart, to sample RNAs from an overlaid tissue section.

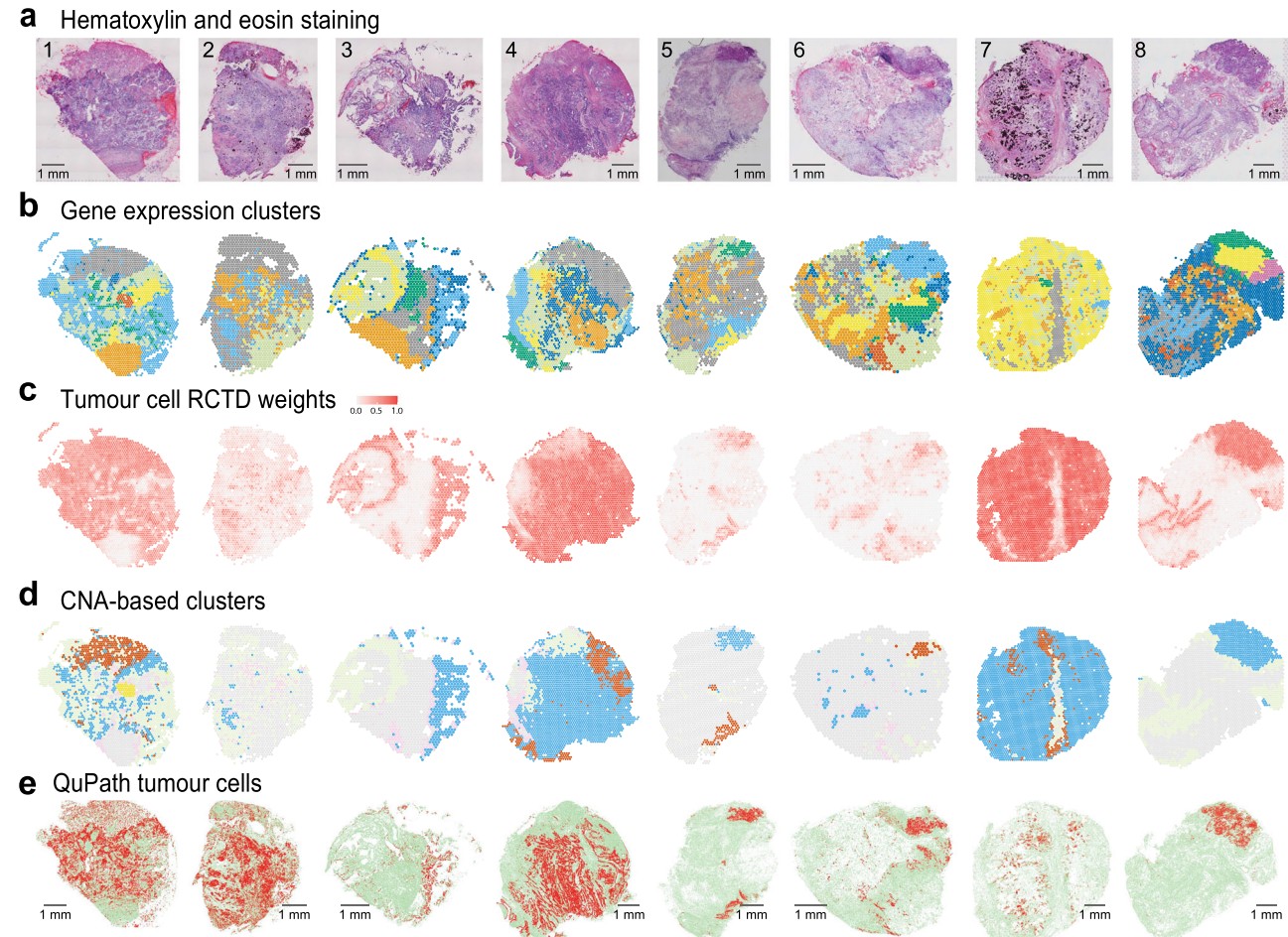

**Fig. 1 | Graphical overview of the Visium data generated for eight HGSOC samples. a** Hematoxylin (blue) and eosin (red) stained tissue sections. Patient IDs are shown for each section. Scale bar = 1 mm. **b** Gene expression-based clustering of the Visium data. Expression profiles for spots are clustered and then mapped back onto the tissue sections. Cluster colours are randomly assigned. **c** Tumour cell enrichment weights calculated using RCTD[27]. Spots with tumour cell enrichment are shown in red. **d** CNA-based clusters. Blue, red, and yellow spots correspond to putative tumour subclones, grey spots are non-malignant regions with RCTD tumour scores <0.15, green and pink correspond to border regions.

**e** Histopathological expert annotation of the tissue sections using QuPath[67]. Red corresponds to malignant cells, green corresponds to stroma. Scale bar = 1 mm. Note that the colours shown in **b** are arbitrary but highlight that unsupervised clustering of the expression data using Seurat (**b**), and clustering of inferCNV profiles (**d**) identified clusters with spatial patterns that largely reflected the morphology shown in **a** and the pathology shown in **e**. Supplementary Fig. S19 shows Sankey diagrams and statistics summarising the relationship between the clusters shown in (**b**) and (**d**).

We employed this technology on sections of primary tumours from eight HGSOC patients to study their cellular composition and tumour microenvironment (Fig. 1a). The tumour samples were collected during interval debulking surgery from HGSOC patients who underwent taxane- and platinum-based neoadjuvant chemotherapy (NACT) (Supplementary Fig. S1, Supplementary Data 1) and included three patients with poor chemotherapy response score (CRS1: patients 1, 7 & 8), three with good response (CRS3: patients 2, 3 & 5) and two with a partial response (CRS2: patients 4 & 6)[26]. The number of Visium spots with data varied across the eight patient samples, ranging from 1501 to 3584 per section, and the median number of genes detected per spot was 2459 (corresponding to 5882 unique molecular identifiers, UMIs) (see **Methods**). Gene expression clustering of the spots in each section identified five to nine clusters per sample (Fig. 1b). The spatial distribution of the identified clusters largely mirrored morphologically distinct regions of the sections seen after haematoxylin and eosin (H&E) staining (Fig. 1a, b). For instance, the clusters shown in grey and orange for patient 1 in Fig. 1b correspond to areas at the top and bottom of the tissue section that are visually distinct from the rest of the section on the H&E staining image (Fig. 1a).

As Visium spots sample transcripts from several cells (typically one to 10), the gene expression profile of a spot can potentially be from a mixture of different cell types. To estimate the cellular composition of each spot, we next applied robust cell type decomposition (RCTD)[27], one of the top-performing methods for cell type deconvolution[28] (**Methods**). Using expression profiles of 12 cell types identified in a scRNA-seq dataset of post NACT HGSOC samples (generated in this study, see **Methods**, Supplementary Fig. S1), RCTD revealed distinct areas with predicted high incidence of tumour cells in each section (Fig. 1c). Positive correlations between RCTD cell type weights indicated co-localisation of some non-malignant cell types (e.g. fibroblast, endothelial cell and myofibroblast scores correlated strongly with each other, Supplementary Fig. S2). There was also a strong anticorrelation between tumour cell weights and B/plasma cell, fibroblast, and macrophage weights (Supplementary Fig. S2). The spatial distribution of cell type weights for non-malignant cell types exhibited notable heterogeneity both between and within samples (Supplementary Fig. S3). Similar results were obtained when we repeated the RCTD spot decomposition using a larger recently published metastatic ovarian cancer scRNA-seq dataset[24]. As expected, the RCTD scores for cell populations common to both scRNA-seq datasets, such as macrophages, endothelial, and T cells, were significantly correlated (Supplementary Fig. S4).

## Tumour subclones with unique CNAs and spatial locations

Copy number alterations (CNAs) are ubiquitous in HGSOC[5]. To predict CNAs for each Visium spot and to cluster spots by similar CNA profiles, we applied inferCNV, which detects differences in average relative expression levels for a sliding window of 101 genes[29]. For each sample we used expression from Visium spots with RCTD tumour cell weights below 0.15 as a normal reference profile. The obtained inferCNV profiles were grouped into clusters, with the number of clusters determined manually based on noticeable variations in residual expression. Clusters with fewer than 10 differentially expressed genes between them were subsequently merged. We then employed Hidden Markov and Bayesian latent mixture modelling within inferCNV to identify high-confidence CNAs within the resulting clusters.

In patient 1, inferCNV predicted multiple large high-confidence CNAs. These included amplifications of specific regions in chromosomes 8, 12, and 20, as well as deletions in parts of chromosomes 6, 17, and 19 (Fig. 2b). In the identified clusters P1.1, P1.2, P1.3, P1.4, and P1.5, inferCNV detected 44, 37, 48, 31, and 5 high-confidence CNAs,

respectively, many of which were shared between clusters (Supplementary Data 2). Notably, clusters P1.1, P1.2, and P1.3 are distinguished by unique CNAs exclusive to each cluster, which suggests that these clusters harbour tumour subclones with different CNAs (Fig. 2c, Supplementary Data 2). In contrast, all the high-confidence CNAs detected in clusters P1.4 and P1.5 were observed in the other clusters, leading us to conclude they were likely border regions containing one of the above clones but with a higher stromal content. Supporting this conclusion, the malignant clusters P1.1, P1.2, and P1.3 had higher average RCTD tumour weights (0.58, 0.57, and 0.48, respectively). It is worth noting that while the CNA patterns observed in P1.4 and P1.5 can be explained by differences in tumour-stroma proportions, the mutually exclusive CNAs observed in P1.1, P1.2 and P1.3 can only be explained by the presence of subclones with distinct CNAs.

In the seven remaining patients, our analysis identified three samples with a single malignant cluster and four samples with two malignant clusters with different CNA profiles (Fig. 1d, Supplementary Fig. S5–11). A summary of overlapping and unique high-confidence CNAs in the eight patients and a comparison to CNAs identified by the TCGA is provided in Supplementary Fig. S12. To confirm that our strategy reliably identified malignant clusters, we carried out a histopathological assessment of the H&E images from the Visium sections, which showed that in most cases tissue areas with low RCTD tumour cell scores corresponded to regions of cells labelled as stroma by a pathologist using QuPath, while high-confidence malignant CNA-based clusters corresponded to the regions of cells called as malignant (Fig. 1e). Notably, repeating the inferCNV analyses using QuPath annotations to identify background spots overlapping morphologically normal cells resulted in substantially worse CNA inference and failed to predict subclones (Supplementary Note 1).

To validate the cluster-specific CNAs predicted in patient 1 for clusters P1.1, P1.2 and P1.3, we performed whole genome amplification and ultra-low-pass whole genome sequencing (WGS) of microdissected regions corresponding to these clusters and a non-malignant control region (**Methods**). For each cluster, several small tissue fragments were isolated from an adjacent section to that profiled by Visium (Fig. 2d). We used ichorCNA[30] to identify large-scale copy number alterations in the DNA extracted from each region. Notably, the CNA profiles of replicates within each cluster were highly correlated with the exception of replicate 1 of P1.1 which had slightly stronger correlation with replicates from the P1.3 and P1.2 clusters (Supplementary Data 3). The averaged ichorCNA signals also strongly correlated with the averaged inferCNV signal across spots within the respective cluster, with Spearman correlation coefficients of 0.66, 0.67, and 0.65 for the clusters P1.1, P1.2, and P1.3, respectively. Furthermore, the WGS confirmed multiple high-confidence CNAs predicted by inferCNV, including those specific to the three clusters and others shared among all malignant clusters. Specifically, the deletion at chromosome 4 in P1.2 was validated across all three replicates, the amplification at chromosome 4 in P1.3 was validated in both replicates, the deletion at chromosome 5 in P1.1 was validated in both replicates, and the amplification at chromosome 19 in P1.3 was confirmed in both replicates (Fig. 2e). Additionally, the amplification of chromosome 12, which was common to all malignant clusters, was also validated (Fig. 2e). Supplementary Fig. S13 shows genome-wide views of the ichorCNA results. Note the observed patterns of validated cluster-specific CNAs cannot be explained by differences in tumour-stroma proportions, thus, we concluded that tissue areas corresponding to the clusters P1.1, P1.2, and P1.3 likely contain tumour subclones that are closely related, sharing several CNAs, but also possessing additional unique CNAs. Figure 2f shows representative areas for these subclones, where hyperchromatic, pleomorphic nuclei, arranged in variably solid, glandular and papillary patterns typical of HGSOC are visible; see Supplementary Fig. S7–10 for other samples.

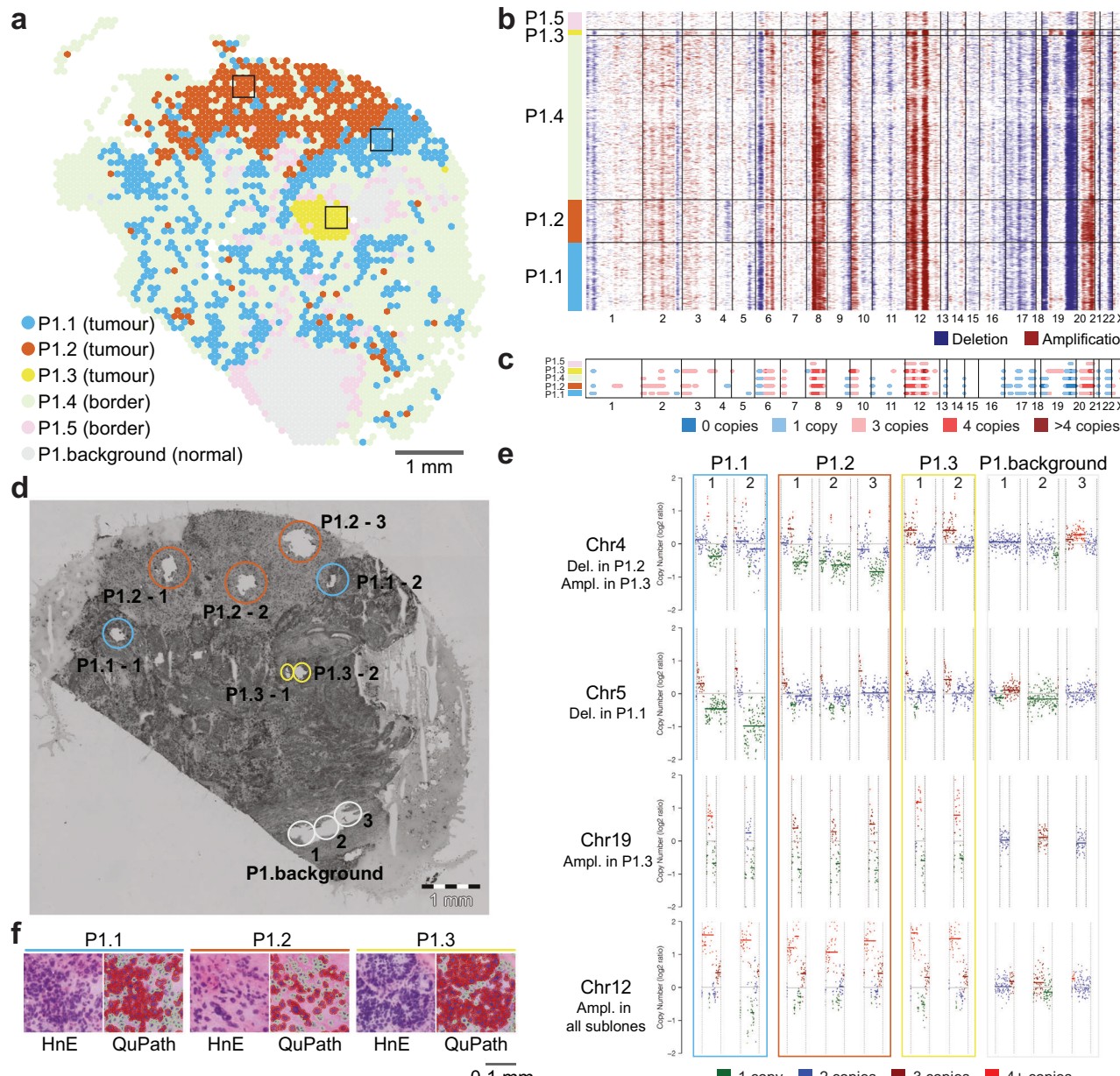

**Fig. 2 | Copy number analysis reveals three tumour subclones with spatially restricted patterns in patient 1. a** Projection of spot clusters identified by inferCNV onto the tissue section. P1.background corresponds to the set of spots used as a reference for inferCNV, P1.1, P1.2, and P1.3 are three putative tumour subclones, P1.4 and P1.5 are probable tumour border clusters. Scale bar = 1 mm. **b** Heatmap generated by inferCNV showing inferred CNA profiles of Visium spots for five clusters. **c** High-confidence CNAs identified by Hidden Markov and Bayesian latent mixture modelling within inferCNV for the five clusters. **d** Adjacent tissue section showing areas collected for low pass whole genome sequencing (WGS).

Colours of ellipses correspond to the colours of clusters in **a**. Scale bar = 1 mm. **e** IchorCNA CNA profiles for selected chromosomes confirming high-confidence CNAs predicted by inferCNV. See Supplementary Fig. S13 for the genome-wide view. Three tumour subclones and non-malignant tissue (P1.background) are shown with two or three replicates each, corresponding to tissue fragments in (**d**). **f** Representative tissue areas for the three subclones. Hematoxylin (blue) and eosin (red) staining (HnE) and histopathological expert annotation using QuPath[67] are shown, red corresponds to malignant cells, green corresponds to stroma. Location of these areas on the tissue is shown by rectangles in (**a**). Scale bar = 0.1 mm.

## Spatial distribution of transcriptionally defined molecular subtypes

HGSOC has previously been classified based on bulk gene expression measurements into four molecular subtypes: mesenchymal, immunoreactive, differentiated, and proliferative[25]. Patients with immunoreactive and differentiated subtypes have been reported to have better outcomes, while those with mesenchymal and proliferative subtypes have poorer outcomes[25,31]. However, the reproducibility and clinical significance of these subtypes remain debated, and no consensus has been reached[32,33]. The Visium data presents a unique opportunity to

investigate these molecular subtypes in a spatial setting where the ratios of tumour and stromal cells vary across a slide from the same patient biopsy; therefore, we employed the Seurat tool[34] and utilised the gene sets identified by PrOTYPE (predictor of high-grade serous ovarian carcinoma molecular subtype)[31] to calculate module scores for each subtype. This analysis revealed variations in subtype signatures across each slide which were different to a whole slide estimate based on all spots, indicating the presence of multiple co-existing subtypes within a Visium section, and that sampling of different regions of the tumour would likely yield different classifications (Fig. 3,

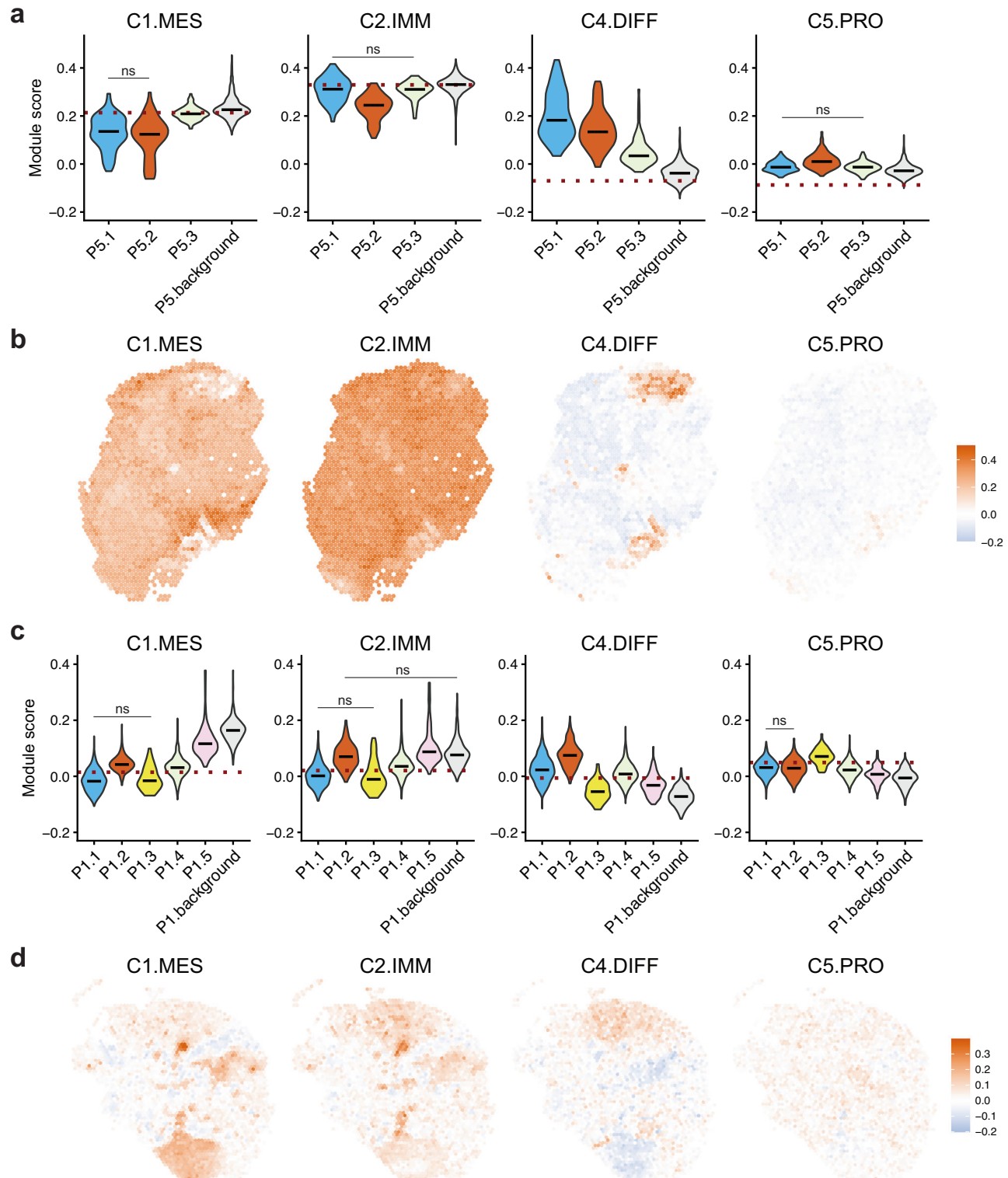

**Fig. 3 | Variations in HGSOC molecular subtype signatures. a** Distribution of Module scores for the four molecular subtype signatures in each of the CNA-based clusters in patient 5. **b** Spatial distribution of Module scores for the four molecular subtype signatures in patient 5. **c** Distribution of Module scores for the four molecular subtype signatures in each of the CNA-based clusters in patient 1. **d** Spatial distribution of Module scores for the four molecular subtype signatures in patient 1. The labels shown correspond to the mesenchymal (C1.MES), immunoreactive (C2.IMM), differentiated (C4.DIF), and proliferative (C5.PRO) subtypes respectively; IMM and DIF are associated with good outcomes while MES and PRO are associated with poor outcomes[25]. For (**a**) and (**c**), black lines are medians, red dotted lines show the value if all spots are combined as a pseudo-bulk. ns and bars indicate seven pairs of clusters where there was no significant difference in the module score; all other pairwise comparisons returned significant results, significance was determined using two-sided Mann-Whitney U test with Benjamini-Hochberg correction and 0.05 threshold. Source Data for panels (**a**) and (**c**) are provided.

Supplementary Fig. S5–7, 9–11). Notably, the relative signal of each subtype varied in regions identified as non-malignant and malignant. For instance, in patient 5, the malignant clusters P5.1 and P5.2 displayed scores associated with a favourable outcome (high differentiated and low mesenchymal), while non-malignant regions exhibited low differentiated and high mesenchymal scores (Fig. 3a, b). Consistently, non-malignant areas in other samples had lower differentiated and higher mesenchymal scores than malignant areas (Fig. 3c, d, Supplementary Fig. S5–7, 9–11). These findings are in agreement with a previous study reporting that varying stroma-to-tumour cell ratios impact on the reproducibility and interpretation of these molecular subtypes[33].

## Gene expression and microenvironment differences between tumour subclones

Our inferCNV analysis detected several malignant clusters with distinct CNAs in five samples, including four samples with two clusters each and one sample with three clusters. The validation by WGS in patient 1 suggests that these clusters likely contain different tumour subclones. To investigate gene expression differences between tumour subclones and variations in the cell types they associate with, we performed pairwise differential gene expression analysis on malignant CNA-based clusters within each patient (total of seven comparisons). This analysis identified between 15 and 233 genes significantly differentially expressed between malignant clusters of each patient (Supplementary Data 4). The combined list of 606 differentially expressed genes was next annotated as tumour cell or stromal cell derived using the scRNA-seq dataset (Supplementary Data 4). Among these genes, 229 (38%) were most highly expressed in tumour cells (many of which fell within amplified genomic regions, Supplementary Data 5), while 60% exhibited the highest expression in other cell types (21 genes in B/plasma cells, 24 in T cells, 37 in myofibroblasts, 41 in endothelial cells, 47 in mesothelial cells, 54 in macrophages, and 141 in fibroblasts). The remaining 2% (12 genes) were not detected in our scRNA-seq dataset.

Next, we compared non-tumour cell infiltration between malignant CNA clusters using RCTD cell type weights. Permutation testing revealed significant within-patient differences across multiple clusters and cell types (Supplementary Fig. S14). As an example, we observed significantly different infiltration by T cells between clusters P1.1 and P1.2, P4.1 and P4.2, as well as between P7.1 and P7.2. Similarly, clusters of patients 5, 6, and 7 showed differential infiltration by B cells, while clusters of patients 1, 4, 5, and 7 displayed differential infiltration by macrophages.

Taken together, our analyses reveal tumour-subclone-intrinsic gene expression differences and evidence that these subclones may be associated with different immune and stromal cell populations. Significantly, genes associated with prognosis and therapy sensitivity were differentially expressed between malignant clusters containing different subclones. For example, the tumour cell expressed genes CD24, CLU, and SLPI[35–37] and the infiltrating cell expressed genes GPNMB, MGP, GPX3, and MFAP4[38–41] have all been previously associated with poor prognosis and chemotherapy resistance in HGSOC. We note in a supplementary analysis comparing malignant spots from the three good response and three poor response samples that genes uniquely expressed by immune cells (B cells and macrophages) were more highly expressed in good response samples and genes expressed in tumour cells were more highly expressed in poor response samples (Supplementary Note 2, Supplementary Data 6).

## Single cell resolution spatial transcriptomics of subclones

We used the NanoString CosMx Spatial Molecular Imaging (SMI) system[42] to spatially examine gene expression at single cell resolution. A 960 gene panel was applied to a serial section of the biopsy profiled by Visium from patient 5. Cellpose[43] was used to identify 39,939 segments corresponding to putative cells. These were filtered to retain segments where at least 100 transcripts were detected and Scrublet[44]

was used to remove segmentation doublets (artifactual fusing of cells due to segmentation failure). This left 21,651 putative singlet cells which were then clustered and annotated. Based on marker genes, 12 distinct cell populations were identified in the CosMx dataset (two tumour cell populations, LYZ+, SPP1+, C1QC+, and CXCL9+ macrophages, CCL5+T cells, IGHA1+ and IGHG1+ plasma cells, COL3A1+ fibroblasts, ENG+ endothelial cells, and KRT17+ epithelial cells; Fig. 4a, b).

We first used the CosMx data to confirm that the two tumour subclones predicted in the malignant clusters P5.1 and P5.2 of the Visium data were present in the serial section. Indeed, two clusters of tumour cells expressing high levels of known HGSOC tumour cell markers and staining positive with a pan-cytokeratin antibody were identified by CosMx SMI in the same positions as clusters P5.1 and P5.2 of the serial section profiled by Visium (top right and bottom right, respectively, Supplementary Fig. S15a, Fig. 4a, c). Furthermore, cells in these two CosMx clusters differ dramatically, with 155 significantly differentially expressed genes (FDR < 0.05, |log₂FC|> 1, Supplementary Data 7), supporting our prediction of different tumour subclones in the corresponding tissue areas.

We next compared genes differentially expressed (FDR < 0.05, |log₂FC|> 1) between P5.1 and P5.2 clusters in Visium data to those differentially expressed between the two tumour cell clusters in the CosMx data. Of 17 genes differentially expressed between P5.1 and P5.2 and present on the CosMx panel, 13 were validated by the CosMx data as differentially expressed between the two tumour cell clusters (Fig. 4d, Supplementary Data 8). This included PIGR in the subclone mapping to P5.1 and PTGS1 in the subclone mapping to P5.2, thus, we henceforth refer to these tumour subclones as PIGR+ tumour cells and PTGS1+ tumour cells, respectively. Of the 13 validated differentially expressed genes, 7 and 6 were consistently identified by both technologies as significantly more highly expressed in the P5.1/PIGR+ or P5.2/PTGS1+ tumour subclones, respectively (Fig. 4d, Supplementary Data 8). For the remaining four genes, two were derived from B/plasma cells (IGKC, IGHA1), and two were differentially expressed between the subclones in the CosMx data but below our two-fold change threshold (KRT17, SLPI).

Importantly, seven genes differentially expressed between P5.1 and P5.2, and initially categorised as stroma-derived based on scRNA-seq data (S100A8, S100A9, CXCL9, INHBB, ADIRF, IGHM, TAGLN), were revealed by the CosMx data to be highly expressed within and differentially expressed between the PIGR+ or PTGS1+ subclones. Remarkably, 132 genes were exclusively identified as significantly differentially expressed between the clones in the CosMx data (Fig. 4d, Supplementary Data 8). This underscores the superior capability of single-cell resolution CosMx spatial data in capturing differentially expressed transcripts intrinsic to the tumour that might have been overlooked by the lower resolution Visium data.

## Subclonal microenvironments at single cell resolution

With the CosMx data we next examined whether the subclones identified in patient 5 preferentially associated with different sets of non-malignant cell types. We first used neighbourhood analyses to survey the proportions of each cell type in the local neighbourhood of PIGR+ and PTGS1+ tumour cells. We employed the Squidpy[45] software to survey the neighbouring cells of each tumour cell at 3 different distances (radii of 110, 340, and 648 pixels) which yielded median numbers of 3, 30 and 100 neighbouring cells, for each of the respective distances. This allowed us to discriminate between cells directly touching the tumour clones, those in the local niche, and those further away. Remarkably, the most common neighbouring cell type was other tumour cells, with a significant proportion of cells neighbouring PTGS1+ cells also being PTGS1+ cells (90%, 80%, and 69% at the respective distances). Similarly, for the PIGR+ subclone, a considerable percentage of neighbours (80%, 63%, and 54% at the respective

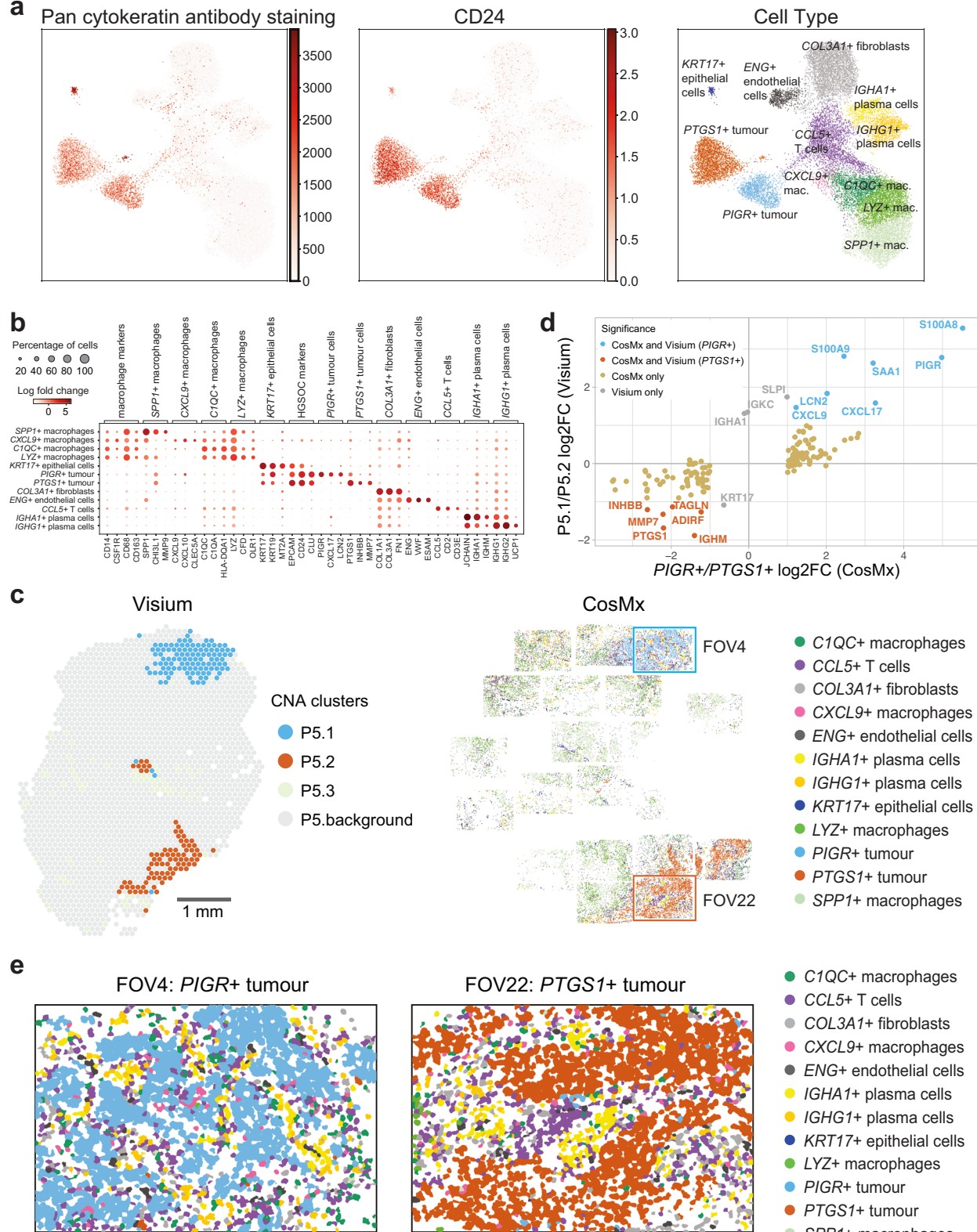

**Fig. 4 | Single cell resolution spatial analysis of HGSOC.** CosMx SMI data shown is from a serial section of that profiled by Visium for patient 5. **a** UMAP representations of the cells identified in the CosMx data. (left) Mean fluorescence intensity within a given cell for pan cytokeratin antibody staining, (centre) Expression of *CD24* - a HGSOC marker, (right) Cell type annotations. **b** Marker genes used for cell cluster annotation. **c** Spatial overview comparing CNA clusters identified by Visium and cell types identified by CosMx SMI. Scale bar = 1 mm. **d** Scatterplot comparing log₂FC of genes identified as differentially expressed in the Visium and CosMx analyses (using two-sided non-parametric Wilcoxon rank sum test in Seurat, with Bonferroni adjustment). The Visium and CosMx data were significantly correlated (Spearman's correlation coefficient of 0.82 and *p*-value < 2.2e−16). **e** Closeup of two CosMx fields of view (FOVs) containing *PIGR* + (left) and *PTGS1* + (right) subclones. The corresponding FOVs are highlighted in **c** in blue and red, respectively.

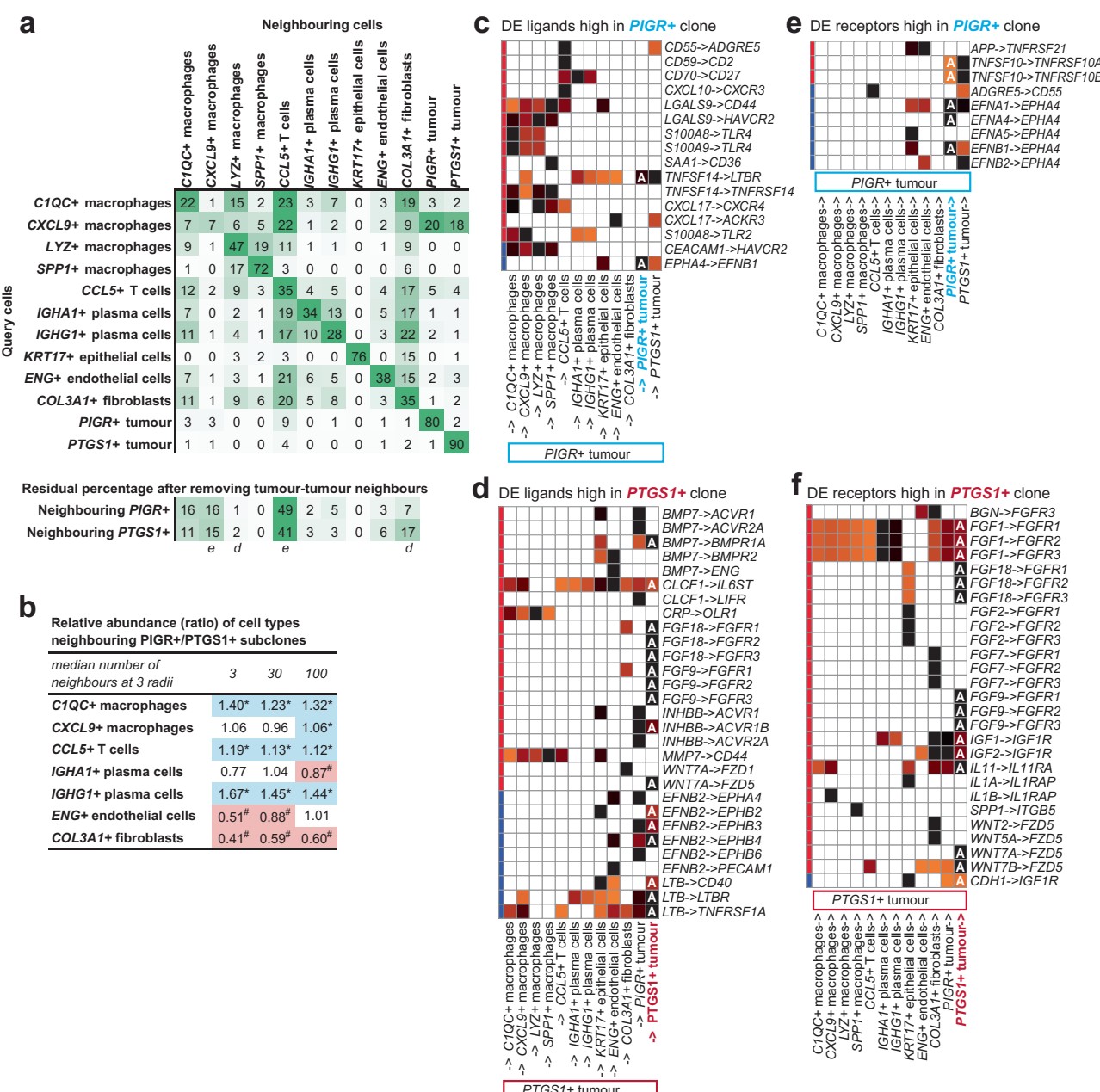

**Fig. 5 | Subclonal microenvironment and ligand and receptor analyses.** Using squidpy we surveyed the cell types neighbouring each cell type at 3 different distances (Supplementary Data 9). **a** Neighbouring cells at the shortest radius of 110 pixels (median of 3 neighbouring cells). Rows indicate query cell type. Columns are neighbouring cell types. Numbers in cells are the percentage across each row. Green indicates maximum value per row. Residual proportions after removing homotypic tumour cell-tumour cell neighbours are also shown. Enriched and depleted cell populations are indicated with *e* and *d*, respectively. **b** Cell neighbourhood analyses showing seven cell populations that each contribute at least 3% of the neighbouring non-tumour cells at three distances yielding medians of 3, 30 and 100 neighbouring cells, respectively. Populations significantly more abundant near the *PIGR*+ and *PTGS1*+ clones are indicated in blue and with *, and in pink with

#, respectively. Ligand-receptors signalling between tumour subclones and cells in their microenvironment involving: **c** Ligands up-regulated in the *PIGR*+ subclone, **d** Ligands up-regulated in the *PTGS1*+ subclone, **e** Receptors up-regulated in the *PIGR*+ subclone, **f** Receptors up-regulated in the *PTGS1*+ subclone. Secreted ligands and plasma membrane ligands are indicated by red and blue bars, respectively. Magma palette used; dark pixels indicate strongest signalling and white indicates no signalling. Autocrine loops are indicated with **a**. Receptors and ligands needed to be detected in ≥ 10% of cells from a cell type to be shown. Note: only 281 of 828 ligands and 229 of 691 receptors in connectomeDB2020[46] are covered on the CosMx platform, thus many subclone-specific signalling events are likely missed in this analysis.

distances) were also *PIGR*+ subclonal cells (Supplementary Data 9; Fig. 5a shows the proportions at radii of 110 pixels). The most common heterotypic neighbour pairs involved *CCL5*+ T-cells and *COL3A1*+ fibroblasts neighbouring other cell types. Prominently, only *CXCL9*+ macrophages were frequently neighboured by tumour cells (Fig. 5a).

To examine the association of the subclones with other cell types, we removed all homotypic interactions between tumour cells and

recalculated the proportions of heterotypic interactions involving the *PTGS1*+ and *PIGR*+ subclones and other cells. Utilising significance testing by cell type label permutations, we consistently observed a significant enrichment of *CCL5*+ T cells and *CXCL9*+ macrophages in close proximity to both subclones across all distances. Conversely, there was a consistent and significant depletion of *LYZ*+ macrophages and *SPP1*+ macrophages near both subclones at all distances (Fig. 5a,

Supplementary Data 9). We then conducted a comparison of heterotypic neighbours of the *PIGR*+ and *PTGS1*+ subclones, with significance determined through cell label permutation. The results indicated that across all three distances, *C1QC*+ macrophages, *CCL5*+ T cells, and *IGHG1*+ plasma cells exhibited a significantly higher likelihood of association with the *PIGR*+ clone. Conversely, *COL3A1*+ fibroblasts showed a significantly higher likelihood of association with the *PTGS1*+ clone (Fig. 5b, Supplementary Data 9). Notably, RCTD analysis of the Visium data found similar associations, more macrophages associated with the P5.1/*PIGR*+ clone and more fibroblasts associating with the P5.2/*PTGS1*+ clone (Supplementary Fig. S14).

### Cell-to-cell communication analysis of tumour subclones

We next used the CosMx data to examine cell-to-cell communication of the *PIGR*+ and *PTGS1*+ subclones identified in patient 5. For these analyses we focused on ligands and receptors that were highly expressed in a subclone (expression ≥ 50% of max expression across all cell types) and differentially expressed between the subclones (FDR < 0.05, $\log_2 FC > 1$, detected in ≥ 10% of cells). In total, 17 ligands and 11 receptors were more highly expressed in the *PIGR*+ clone and 16 ligands and 14 receptors were more highly expressed in the *PTGS1*+ clone.

To predict how these subclone-specific differences in ligand and receptor expression levels might alter their interactions with cells expressing cognate receptors and ligands in their microenvironment, we used NATMI and the literature-supported ligand and receptor database connectomeDB2020[46]. This revealed that ligands from the *PIGR*+ subclone predominantly bind receptors on macrophages and T cells, while those from the *PTGS1*+ clone predominantly bind receptors on epithelial cells and endothelial cells (Fig. 5c, d). Examples of *PIGR*+ subclone derived ligands predicted to signal to cells more abundant near the *PIGR*+ clone include *S100A8* and *S100A9* signalling to *C1QC*+ macrophages, and *CD55*, *CD59* and *CXCL10* signalling to *CCL5*+ T cells (Fig. 5c). Similarly, the *PTGS1*+ subclone derived ligands *EFNB2*, *CLCF1*, and *BMP7* are predicted to signal to *ENG*+ endothelial cells which were more abundant close to *PTGS1*+ cells (Fig. 5b, d).

Surprisingly, multiple subclone specific autocrine loops were also predicted. This was achieved by up-regulation of receptors, ligands or both (Fig. 5c–f). For the *PIGR*+ clone, autocrine loops involving *EFNA1/EFNA4/EFNB1->EPHA4*, *TNFSF10->TNFRSF10A/TNFRSF10B*, and *TNFSF14->LTBR* were predicted (Fig. 5c, e). Similarly for the *PTGS1*+ clone, autocrine loops involving *BMP7->BMPR1A*, *CDH1->IGF1R*, *CLCF1->IL6ST*, *EFNB2->EPHB2/EPHB3/EPHB4*, *FGF1/FGF18/FGF9 ->FGFR1/FGFR2/FGFR3*, *IGF1/IGF2->IGF1R*, *IL11->IL11RA*, *INHBB->ACVR1B*, *LTB->CD40/LTBR/TNFRSF1A*, and *WNT7A/WNT7B ->FZD5* were observed (Fig. 5d, f).

Lastly, as CosMx data was only available for one patient, we used the Visium data to calculate correlations between tumour cell expressed ligands and RCTD-estimated cellular infiltrates across all eight samples. This identified multiple tumour ligands, whose expression was correlated with cellular infiltrates, for further exploration in HGSOC (e.g., expression of *CXCL10* correlated with RCTD scores for *CXCR3*-expressing T cells; see Supplementary Note 3, Supplementary Data 10).

### Discussion

The importance of clonal heterogeneity in HGSOC cannot be overstated, as it directly impacts recurrence, relapse, and therapy response[8,24,47]. For instance, intratumoural clonal heterogeneity[8] and expansion of pre-existing clones[24,47] have been shown to be associated with reduced survival after platinum based chemotherapy. Disappointingly, a recent report also found evidence of patients with subclones that are likely to be resistant to PARP inhibition[47]. Using Visium spatial transcriptomics, we investigated intratumoural heterogeneity in HGSOC samples from eight patients. This allowed us to

study the spatial relationships between tumour subclones, the ligands and receptors they express, and the cells in their microenvironments.

Building upon prior reports that most HGSOCs are polyclonal[14,15], our spatially resolved copy number inference revealed multiple subclones with different CNAs in five of the eight samples. This mirrors findings from a recent report of spatially inferred CNAs[48]. Our workflow also identified CNAs in two independent cohorts[49,50] of patients with spatial transcriptomics data and for one cohort[49] identified two samples from short term survivors with subclonal CNAs (Supplementary Fig. S16). Subclones were not observed in the second cohort[50], however, the samples studied were approximately seven times smaller, making it unlikely they would sample multiple clones.

Remarkably, the cellular composition and histology of tumour regions containing different subclones varied significantly (Fig. 2a, d, Supplementary Fig. S14). In the case of the two subclones present in patient 5, single-cell resolution spatial transcriptomics with CosMx SMI revealed both were highly infiltrated by *CCL5*+T cells, *C1QC*+ and *CXCL9*+ macrophages. Notably, these cells formed clusters resembling tertiary lymphoid structures which have been reported as predictive of prognosis in HGSOC[51] (Fig. 4e). The CosMx data also revealed that *C1QC*+ macrophages, *CCL5*+ T cells, and *IGHG1*+ plasma cells were more abundant near one tumour subclone while *COL3A1*+ fibroblasts were more abundant near the other.

Differential expression analysis between subclones revealed ligands and receptors were over-represented. This raises the possibility that tumour subclones may shape the composition of their local microenvironments. In the case of patient 5, CosMx data predicted multiple plausible subclone-specific signalling edges. For example, the *PIGR*+ subclone expressed multiple ligands, including *S100A8, SAA1*, and *CXCL10*, which were predicted to signal to *CXCL9*+, *C1QC*+, *LYZ*+ macrophages*, SPP1*+ macrophages and *CCL5*+T cells, respectively (Fig. 5c). Amongst these, *S100A8* was identified as a potential ligand driving higher association of *C1QC*+ macrophages with the *PIGR*+ clone. Others have previously attempted to relate tumour-expressed ligands to differences in infiltration patterns[52–54]; however, to our knowledge, none have reported that subclones expressing different ligands may modulate their local tumour microenvironment.

Analysis of subclonally perturbed ligands and receptors from CosMx SMI also revealed subclone-specific autocrine loops, several of which are clinically relevant (Fig. 5c–f). For example, multiple fibroblast growth factors and their receptors were up-regulated in one clone to create multiple *FGF - > FGFR* mediated autocrine loops. These include *FGF18* which is a marker for poor prognosis[55], *FGF9* which is induced by NACT and correlates with time to recurrence[56], and both *FGFR1* and *FGFR2* which are implicated in cisplatin resistant HGSOC[57]. In the same subclone we also observed an autocrine loop involving *WNT7A -> FZD5* which has been previously linked to HGSOC tumour growth and progression[58]. Our study is not the first to report autocrine loops in HGSOC[59,60]. However, the upregulation of different sets of autocrine loops in each subclone suggests tumour heterogeneity facilitates evolutionary exploration of this space and reinforces the importance of autocrine signalling to tumour fitness proposed almost 45 years ago[61,62].

In closing, our findings underscore the pervasive polyclonality inherent to HGSOC. Notably, multiple genes associated with poor prognosis were differentially expressed between subclones in our data. This polyclonality poses challenges for molecularly targeted personalised therapies[63], necessitating strategies that address vulnerabilities common to all clones within a patient or employ combination therapies targeting individual clones. As immune infiltration is associated with a good treatment response[64,65], the study of subclones with different microenvironments represents an opportunity to uncover ligands that potentially influence the formation of anti-tumour microenvironments such as tertiary lymphoid structures. Harnessing

these identified ligands for therapeutic interventions holds promise in guiding tumours towards more positive outcomes.

## Methods

### Ovarian tumour samples and consent

The study was approved by the St John of God Health Care (SJGHC), The University of Western Australia (UWA) and Curtin University Human Research Ethics Committees (approval numbers #1217 and RA/4/20/5784). All participants were given information about the study and provided written informed consent before enrolment.

High-grade serous ovarian tumours from 10 patients diagnosed with stage III-IV cancers were included in this study. For three patients, samples were profiled using both Visium and scRNA-seq, see Supplementary Data 1. Patients were treated with 3-6 cycles of platinum-based chemotherapy. All tumour samples were derived from ovarian sites during interval debulking surgery. Fresh tumours were either collected in RPMI (ThermoFisher Scientific) supplemented with penicillin and streptomycin (Sigma) for single-cell dissociation or immediately snap-frozen and stored in −80 °C tumour bank until retrieved for Visium experiments.

The chemotherapy response scores (CRS, 3 tier) for each patient were determined as previously described[66] by assessing the largest (macroscopic) omental tumour deposit for features of regression based on the following: score 1 (no/minimal tumour response), score 2 (partial tumour response) and score 3 (complete/near complete response, cell groups measuring <2 mm each, or no residual tumour)[26].

### Statistics & reproducibility

No statistical method was used to predetermine sample size. No data were excluded from the analyses, with the exception of low quality cell segments and spots filtered out as described below in the sections on the Visium and CosMx analyses. The experiments were not randomised. The investigators were not blinded to allocation during experiments and outcome assessment. Statistical tests used in the manuscript are described in the main text and in the methods below.

### Histopathological assessment

Whole-slide H&E images from Visium tissue sections were analysed using QuPath[67] version 0.4.3. To classify tumour and stroma cells, whole-tissue regions of interest were selected with positive cell detection based on hematoxylin nuclei staining. Segmentation of tumour and stroma areas were then performed using QuPath's train object classifier command. Representative training areas were annotated by an experienced pathologist using the annotation tools to classify tumour area (red) and stroma area (green). QuPath algorithm used available features to train the classifier to provide optimal classification performance. A minimum of 5 training rounds were performed to ensure adequate segmentation of the tumour and stroma regions. Once satisfactory, the classifier was applied to the whole image. The images with tissue classification overlay were then exported.

### Single-cell suspension for scRNA-seq

HGSOC tumours were dissociated using the tumour dissociation kit 2, human, from Miltenyi Biotec [130-095-929] as per manufacturers' instructions. Tumour tissue (0.2–1 g) was cut into small pieces of 2–4 mm and placed into a gentleMACS C-tube [Miltenyi Biotech; 130-093-237] containing the enzyme mix from the kit. The tube was then placed onto the gentleMACS octo dissociator (Miltenyi Biotech) and processed using the 37C_h_TDK_1 program with the associated incubation times indicated in the protocol. Complete tissue dissociation was confirmed by the absence of visible tissue chunks. The resulting tumour homogenate was filtered using a 70-μm MACS SmartStrainer and washed with RPMI. Cell suspension was further filtered through 40-μm strainers to remove cell clumps. The viability was assessed by

ReadyProbe Cell Viability Imaging Kit (ThermoFisher Scientific) to ensure the viability was >90%.

### Single-cell RNA-seq profiling

Cryostored cells were rapidly thawed in a water bath set at 37 °C. 1 mL of Media (RPMI1640 + 10%FBS) was then added to the cells, which were then mixed and transferred to a 15 mL falcon tube. The cryovial was then rinsed with another 1 mL of media which was subsequently added to the 15 mL falcon tube in a dropwise fashion. 7 mL of media was then added to the falcon tube dropwise using a serological pipette. The cells were then centrifuged at 300 $g$ for 5 minutes. The supernatant was removed, leaving behind 1 mL of media, and another 1 mL of media was added and the cells were resuspended. 2 mL of DPBS + 0.04%BSA was then added to the cells, followed by another centrifugation at 300 $g$ for 5 minutes. The supernatant was then removed and the cells were resuspended in 1 mL of DPBS + 0.04%BSA and then subsequently run through a 40 uM filter. Cells were then counted and viability was checked using the Countess II automated cell counter and the readyprobes red/blue viability kit (Thermo Fisher Scientific). Libraries were prepared in accordance with the protocol for 10x Chromium Single Cell 3' v2 (10x Genomics). Sequencing was performed on a NovaSeq 6000 (Illumina).

### Visium spatial transcriptomic profiling

Frozen tissue fragments were embedded in Tissue-Tek O.C.T. Compound (25608-930, VWR) according to the Visium Spatial Protocols – Tissue Preparation Guide (CG000240 Rev A, 10x Genomics) and stored immediately at −80 °C until further use. Hematoxylin and eosin staining of 10 μm cryosections from each O.C.T. block were assessed by a pathologist to confirm tissue type and tumour content. Samples with adequate tumour content were selected for use in the gene expression workflow.

To assess the quality of the selected tissue blocks, RNA was isolated from serial sectioned tissues totalling 80 μM thickness and its RNA integrity number (RIN) was calculated using the Agilent 4200 TapeStation system. Samples which had a RIN ≥ 7 were considered good quality and selected to proceed with the experiment. Each Visium Spatial Gene Expression Slide (2000233, 10x Genomics) was used to analyse up to four tissue samples, i.e. one section per sample block. Of the four samples, one block was randomly selected for tissue optimisation using a Visium Spatial Tissue Optimisation Slide (3000394, 10X Genomics). Serial tissue sections at 10 μM thickness were placed on seven capture squares of the pre-chilled tissue optimisation slide with the remaining square left empty. Different tissue permeabilisation times were tested on 6 of the sections at 10-minute intervals to a maximum of 60 minutes. The remaining tissue section represented a negative control for permeabilisation while the empty well served as a positive control to which a reference RNA was added (QS064, Life Technologies). The optimal permeabilisation time point was 30 minutes and was therefore used as the permeabilisation time on the gene expression samples.

A Nikon Eclipse Ni-U microscope with a 10x objective in large scale imaging mode (Nikon, NIS-Elements AR 5.21.00) was used to take brightfield images of the Visium Spatial Tissue Optimisation and Gene Expression slides. The same settings were used to collect fluorescent images of the optimisation slides via a Texas Red HYQ filter cube at 1.5 seconds exposure time. Images were automatically stitched via blending with a 10% tile overlap. Original files were saved as the default '.ND2' format and exported to '.tiff' or '.jpeg' using NIS-Elements AR or ImageJ, respectively.

Libraries were prepared according to the Visium Gene Expression User Guide (CG000239, Rev A, 10X Genomics) and pooled to a final library concentration of 1.8 nM. The samples were loaded on a NovaSeq 6000 System (Illumina) using NovaSeq 6000 SP Reagent kit (200 cycles, 20040326, Illumina) and sequenced at a depth of

approximately 150 M reads per sample. The read protocol was set as the following: read 1 at 28 cycles, i7 index read at 10 cycles, i5 index read at 10 cycles and read 2 at 120 cycles.

Manual image alignment and spot selection of the H&E brightfield images was performed in the Loupe Browser.

## Ultra-low-pass DNA sequencing

Frozen tissue was sectioned (10 μm) and mounted to standard superfrost slides, methanol fixed, stained with hematoxylin and eosin, and scanned on a CellCelector (ALS). Using the CellCelector, the long edge of a 150 μm glass capillary was used to mechanically scrape small tissue sections from the slide which were aspirated and deposited in 1 μL of PBS (10 mM Phosphate, 2.68 mM Potassium Chloride, 140 mM Sodium Chloride, 18912014, Thermo Fisher Scientific) in 0.2 mL PCR tubes (Eppendorf). Tissue sections were subjected to whole genome amplification using the Ampli1 WGA Kit (Silicon Biosystems) according to the manufacturer's instructions. Following amplification, 400 bp sequencing libraries were constructed using the Ampli1 Low-Pass Whole Genome Sequencing Kit for Ion Torrent (Silicon Biosystems) according to the manufacturer's instructions. Libraries were diluted to 50 pM, loaded into an Ion Chef for template preparation and loading into an Ion 530 chip, and then sequenced for 525 flows on an Ion S5 (Thermo Fisher Scientific). Sequencing data was aligned to hg38 and indexed using Torrent Server (V 5.16) with depths ranging from 0.1 to 0.3x. Following alignment and indexing,.wig files were generated using readCounter in 1 Mb windows from HMM Copy Utils[68]. IchorCNA (v0.2.0)[30] was used to detect somatic copy number alterations with 1 Mb bins and the run parameters set to --ploidy c(2,3,4), --normal c(0.05), --includeHOMD False, --chrTrain c(1:22), and --estimateScPrevalence False. An unrelated non malignant tongue tissue biopsy was used to construct a panel of normals using IchorCNA's createPanelOfNormals.R script.

## NanoString CosMx SMI profiling

CosMx SMI profiling was carried out on a serial section of tumour from patient 5 adjacent to that profiled by Visium. The fresh frozen section was transferred to a slide and shipped to NanoString for profiling as part of their early technology access program (TAP).

**CosMx SMI sample preparation.** Fresh frozen (FF) tissue sections were prepared for CosMx SMI profiling as described in He et al.[42]. Briefly, five-micron tissue sections on VWR Superfrost Plus Micro slides (cat# 48311-703) were stored at −80 °C, then heated to remove excess moisture using a hair dryer on high for 5 min and baked at 37 °C for 30 min to improve tissue adherence to the slide. Tissues were fixed by incubation with 10% neutral buffered formalin (NBF) diluted in phosphate buffered saline (PBS) for 30 min at 4 °C, then washed three times with 1x PBS for 2 min each. Following fixation, tissues were successively dehydrated in 50% ethanol for 5 min, 70% ethanol for 5 min, and 100% ethanol twice for 5 min each. Tissues were prepared for in-situ hybridization (ISH) by heat-induced epitope retrieval (HIER) at 100 °C for 8 min using ER1 epitope retrieval buffer (Leica Biosystems product, citrate-based, pH 6.0) in a pressure cooker.

After HIER, the tissue sections were digested with 5 μg/ml Proteinase K diluted in ACD RNAscope® LS Protease IV at room temperature for 30 minutes. Slides were washed twice with 1x PBS and incubated in 0.0004% diluted fiducials (Bangs Laboratory) in 2X SSCT (2X saline sodium citrate, 0.001% Tween-20) solution for 5 min at room temperature in the dark. Excess fiducials were rinsed from the slides with 1X phosphate buffered saline (PBS) and tissue sections were fixed with 10% neutral buffered formalin (NBF) for 1 min at room temperature. Fixed samples were rinsed twice with Tris-glycine buffer (0.1 M glycine, 0.1 M Tris-base in DEPC H₂O) and once with 1X PBS for 5 min each before blocking with 100 mM N-succinimidyl (acetylthio) acetate (NHS-acetate, ThermoFisher) in NHS-acetate buffer (0.1 M NaP, 0.1%

Tween PH 8 in DEPC H₂O) for 15 min at room temperature. The sections were then rinsed with 2X saline sodium citrate (SSC) for 5 min and an Adhesive SecureSeal Hybridization Chamber (Grace Bio-Labs) was placed over the tissue.

NanoString ISH probes were prepared by incubation at 95 °C for 2 min and placed on ice, and the ISH probe mix (1 nM 980 plex ISH probe, 10 nM Attenuation probes, 1X Buffer R, 0.1 U/μL SUPERase•In™ [Thermofisher] in DEPC H₂O) was pipetted into the hybridization chamber. The hybridization chamber was sealed to prevent evaporation, and hybridization was performed at 37 °C overnight. Tissue sections were rinsed of excess probes in 2X SSCT for 1 min and washed twice in 50% formamide (VWR) in 2X SSC at 37 °C for 25 min, then twice with 2X SSC for 2 min at room temperature and blocked with 100 mM NHS-acetate in the dark for 15 min. A custom-made flow cell was affixed to the slide in preparation for loading onto the CosMx SMI instrument.

**CosMx SMI instrument run.** RNA target readout on the CosMx SMI instrument was performed as described in He et al.[42]. Briefly, the assembled flow cell was loaded onto the instrument and Reporter Wash Buffer was flowed to remove air bubbles. A preview scan of the entire flow cell was taken, and 20 fields of view (FOVs) were placed on the tissue to match regions of interest identified by H&E staining of an adjacent serial section. RNA readout began by flowing 100 μl of Reporter Pool 1 into the flow cell and incubation for 15 min. Reporter Wash Buffer (1 mL) was flowed to wash unbound reporter probes, and Imaging Buffer was added to the flow cell for imaging. Eight Z-stack images (0.8 μm step size) for each FOV were acquired, and photo-cleavable linkers on the fluorophores of the reporter probes were released by UV illumination and washed with Strip Wash buffer. The fluidic and imaging procedure was repeated for the 16 reporter pools, and the 16 rounds of reporter hybridization-imaging were repeated multiple times to increase RNA detection sensitivity.

After RNA readout, the tissue samples were incubated with a 4-fluorophore-conjugated antibody cocktail against CD298/B2M (488 nm), PanCK (532 nm), CD45 (594 nm), and CD3 (647 nm) proteins and DAPI stain in the CosMx SMI instrument for 2 h. After unbound antibodies and DAPI stain were washed with Reporter Wash Buffer, Imaging Buffer was added to the flow cell and eight Z-stack images for the 5 channels (4 antibodies and DAPI) were captured.

**Image processing and feature extraction.** Raw image processing and feature extraction were performed using an in-house CosMx SMI data processing pipeline[42] which includes registration, feature detection, and localisation. 3D rigid image registration was made using fiducials embedded in the samples matched with the fixed image reference established at the beginning of the CosMx SMI run to correct for any shift. Secondly, RNA image analysis algorithm was used to identify reporter signature locations in X, Y, and Z axes along with the assigned confidence. The reporter signature locations and the associated features were collated into a single list. Lastly, the XYZ location information of individual target transcript was extracted and recorded in a table by secondary analysis algorithm, as described in He et al.[42].

**Cell segmentation.** The Z-stack images of immunostaining + DAPI were used for drawing cell boundaries on the samples. A cell segmentation pipeline using the machine learning algorithm Cellpose[43] was used to accurately assign transcripts to cell locations and subcellular compartments. The transcript profile of individual cells was generated by combining target transcript location and cell segmentation boundaries. Based on the Cellpose segmentation results, we initially obtained 39,939 cells. After excluding cells with less than 100 transcripts, 26,895 cells were kept for downstream analysis. Across all FOVs, 9,915,852 transcripts were detected; 80.97% of these were detected in cells with ≥ 100 transcripts. Cell segmentation doublets (artifactual segments containing merged signals from adjacent cells)

were removed using Scrublet[44] which left 21,651 which were used for gene expression clustering, differential expression and cell-to-cell communication analyses.

## Single-cell RNA-seq data analysis

**scRNA-seq data processing.** FASTQ files were processed using Cell Ranger 3.0.2 with refdata-cellranger-GRCh38-3.0.0 reference. Raw gene-barcode matrices from Cell Ranger output were used for downstream processing. Cells were distinguished from background noise using EmptyDrops[69]. Genes detected in a minimum of 3 cells were retained; cells with at least 500 genes, at least 1000 UMIs and under 15% of mitochondrial reads were retained. Seurat v3 was used for sample normalisation (SCTransform, mitochondrial and ribosomal mapping percentage were regressed out), integration (anchor-based method with 3000 variable genes), dimensionality reduction and clustering (using first 30 principal components), and differential expression analysis (Wilcoxon test)[70]. The integrated dataset contains 17,192 cells in total, with a median of 775 genes and 2002 UMIs per cell.

**Inferring cell identity.** To infer cell identities, we first performed a reference-based annotation using scMatch[71] and a reference dataset from Olbrecht et al. ovarian cancer samples[21]. To construct the reference dataset, we obtained gene counts and cell types reported in Olbrecht et al.[21]; counts were normalised to cell library size and averaged within each cell type to derive reference vectors for scMatch. We then used scMatch with parameters --testMethod s --keepZeros n to label each individual cell with the closest cell type identity from the reference dataset. This resulted in seven major cell types: tumour cells, fibroblasts, ovarian stroma, endothelial cells, monocytes, T cells, and B cells. We then examined expression of differential expressed and cell type marker genes in these seven cell types and based on this relabelled two of them to better reflect the cell identity (B cells to B/plasma cells based on the expression of *IGHG1, IGHG3, JCHAIN*; monocytes to macrophages based on the expression of *CCL3, CXCL8, HLA-DRA*) (Supplementary Fig. S17a, b).

Cells labelled as ovarian stroma and fibroblasts formed multiple visually distinct cell groups. To explore possible subtypes, we extracted these cells from the dataset and reran principal component and clustering analysis for this subset. We identified ten clusters (Supplementary Fig. S17c, d). Cell identities were assigned to these clusters based on their specific differentially expressed genes and cell type gene markers.

**Cluster 1** was characterised by high expression of contractile genes including *TAGLN, ACTA2, MYL9, MYH11, PLN* and was labelled Myofibroblasts. **Cluster 6** was labelled Mesothelial cells based on *CALB2, MSLN, SLPI, KRT8, KRT18* expression, as per Qian et al.[72] and Olbrecht et al.[21]. **Cluster 3** showed high expression of *COL1A1, COL1A2, COL3A1, SPARC, FN1* and was labelled Fibro5 (*FN1, COL3A1*). In **Cluster 8,** *CFD* and *RAMP1* were the top DEGs, and the cluster was labelled Fibro3 (*RAMP1, CFD*) - these cells might correspond to adipogenic fibroblasts, as per Qian et al.[72] and Olbrecht et al.[21]. **Cluster 7** specifically overexpressed *CCL2* and we labelled it Fibro4 (*CCL2*). **Cluster 2** did not have genes strongly overexpressed with $\log_e$FC > 1, but overexpressed *LUM, DCN, GSN* with $\log_e$FC >= 0.5 and was labelled Fibro2 (*RBP1, DCN*). **Cluster 0** and **Cluster 5** were labelled Fibro1 (*EIF4A3, STAR*).

**Cluster 4** showed high expression of stress response-related genes, such as *HSPA6, HSPA1B, DNAJB1, HSPA1A*, hence we assumed it corresponded to cells showing strong stress response and removed it from the downstream analysis. Finally, **Cluster 9** had high expression of genes normally expressed in immune cells, such as *B2M, CCL5, HLA-A, HLA-B, CXCR4*, and these cells co-clustered with T cells in the superset, hence, we concluded this cluster corresponded to doublets and removed it from the downstream analysis.

Final fine-grain annotations for the scRNA-seq dataset are shown in Supplementary Fig. S18 and cell type labels for each of the cells are available in Supplementary Data 11.

## Visium data processing

FASTQ files were processed using Space Ranger 1.0.0 with GRCh38-3.0.0 reference in the manual alignment mode. Filtered gene-barcode matrices from Space Ranger output were used for downstream analyses; barcodes with less than 400 genes were excluded. Seurat v3 was used for sample normalisation (SCTransform, mitochondrial and ribosomal mapping percentage were regressed out), individual sample clustering (using first 30 principal components), integration (anchor-based method with 3000 variable genes), dimensionality reduction and clustering (using first 30 principal components), and differential expression analysis (Wilcoxon test)[70]. Cellular composition of each spot was deconvolved using robust cell type decomposition (RCTD)[27].

## CNA inference

InferCNV[29] was run for each sample independently using Visium spots with RCTD tumour cell weights below 0.15 as a background. The following parameters were used to generate the CNA heatmaps: *cutoff = 0.1, denoise = T, HMM = F*. Spots in each sample were clustered using the default parameters and the dendrogram was split into clusters with visually distinctive CNA profiles. Then inferCNV was run again with *HMM = T* to identify high-confidence CNAs across the clusters.

## NanoString CosMx data analysis

By performing cell segmentation on CosMx data using Cellpose[43], we generated gene expression profiles for identified cells. With squidpy (v1.2.1)[45] and scanpy (v1.9.3)[73], unsupervised graph-based clustering of cells (Leiden algorithm, resolution parameter set to 2) identified 21 distinct clusters within the CosMx SP5 dataset of 20 fields of view (FOVs). Marker genes were identified and used to manually annotate these clusters, which were collapsed into 12 major cell types.

We utilised the generic coordinates of cells to construct a neighbourhood graph based on the distance between two cells using the squidpy library. If the distance is shorter than the given radius, then these two cells are neighbours to each other and there is an edge connecting these two cells in the neighbourhood graph. This approach allowed us to identify cells that were physically close within each field of view or across the entire sample. Subsequently, we generated a neighbour count matrix by tallying the number of neighbours from every cell type/cluster for each cell type/cluster. Each row of the matrix represents the neighbour counts of all cell types/clusters associated with a specific cell type/cluster. To facilitate comparison, we normalised the counts by the total counts in each row, thereby expressing the values as percentages that indicate the proportion of neighbours from each cell type/cluster for a given cell type/cluster.

For neighbour enrichment and depletion testing, positions of tumour cells were fixed and positions of cells of each other cell type were randomised independently 1,000 times. For ratio significance testing, positions of tumour cells were fixed, neighbouring cells were selected for each of 3 radii and their labels were permuted 1,000 times. Monte Carlo procedure was used to calculate empirical *P* values which were then corrected for multiple testing using Benjamini-Hochberg procedure. Threshold of 0.05 was used to define significance.

## Cell-to-cell communication analyses

We used connectomeDB2020[46] supplemented with the following updated ligand-receptor pairs: *ANXA2-TLR4, CD24-SELE, CD24-L1CAM, HMGB1-CD24, CXCL10-ACKR2, CXCL17-CXCR4, CXCL17-ACKR3, CD24-SIGLEC10, S100A8-TLR2* and removal of interactions involving *HSP90AA1*. Ligands and receptors detected in less than 10% of cells of a given cluster were excluded from the analysis.

## Reporting summary

Further information on research design is available in the Nature Portfolio Reporting Summary linked to this article.

## Data availability

The scRNA-seq and Visium data are available from the Gene Expression Omnibus (GEO) repository with the primary accession code GSE211956. The NanoString CosMx data and code are available from Zenodo (https://zenodo.org/records/10048057)[74]. Source data for plots in Fig. 3 are provided with this paper. The two public scRNA-seq datasets reused in this study are available in the Gene Expression Omnibus (GEO) database under accession code GSE165897[24] and from http://blueprint.lambrechtslab.org[21]. The two public spatial transcriptomics datasets reused in this study are available in the Gene Expression Omnibus (GEO) database under accession code GSE189843[50] and from CodeOcean (https://codeocean.com/capsule/1912679/tree/v1)[49]. The low pass whole genome sequence data is deposited at dbGaP under accession code phs003561.v1.p1. The WGS data is available under restricted access to protect the donor's privacy. Access to the data for cancer research purposes is via dbGaP and can be requested by permanent employees of an institution at a level equivalent to a tenure-track professor or senior scientist with laboratory administration and oversight responsibilities. The requests are managed by the Data Access Committee of the NCI, and after approval, access is granted for 12 months. Source data are provided with this paper.

## Code availability

Code for the CosMx analyses is available from Zenodo (https://zenodo.org/records/10048057)[74]. Visium and scRNA-seq data analyses used standard tools that are cited in the manuscript.

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

## Acknowledgements

We would like to acknowledge the patients who participated in this study. This research was carried out during the tenure of an Early Career Investigator Grant from Cancer Council Western Australia to L.dK. It was also supported by a collaborative cancer research grant to A.R.R.F. provided by the Cancer Research Trust (Enabling advanced single-cell cancer genomics in Western Australia), and an enabling grant from Cancer Council of Western Australia. R.H. was supported by an Australian Government Research Training Program (RTP) Scholarship. A.RR.F. was supported by funds raised by the MACA Ride to Conquer Cancer, a Senior Cancer Research Fellowship from the Cancer Research Trust and by an Australian National Health and Medical Research Council Fellowship APP1154524 and Investigator grant APP2025225. Analysis was made possible with computational resources provided by the

Pawsey Supercomputing Centre with funding from the Australian Government and the Government of Western Australia. Genomic data was generated at the Australian Cancer Research Foundation Centre for Advanced Cancer Genomics. Thank you to Joost Lesterhuis for useful comments on the manuscript and Marshall Feterl at Nanostring for assistance with the CosMx technology access program.

## Author contributions

The project was conceived by A.R.R.F., L.dK., P.A.C., Y.Yu. Y.Yeow. prepared the single cell suspensions. M.J., K.P.W.H., S.F., D.O'M., Y.Yeow generated scRNA-seq libraries. L.dK generated the Visium libraries. M.J. did the sequencing. Y.K., R.J., T.S.G., E.K. generated the NanoString CosMx data. E.D. carried out computational analyses and data interpretation of Visium and scRNA-seq datasets. R.H. and A.R.R.F. analysed the NanoString CosMx data. A.T. and M.B. did the pathological annotations. G.R.K.A.M., S.S. carried out the surgical collections. S.B. handled the ethics. A.B. and E.G. did the low pass WGS and ichorCNA analysis. The manuscript was written by E.D. and A.R.R.F. with input from Y.Yu, P.A.C., R.H. and all authors.

## Competing interests

Paul A. Cohen reports speakers' honoraria from AstraZeneca and Seqirus, and consultancy fees and stock in Clinic IQ Pty Ltd. Emily Killingbeck is an employee of NanoString Technologies and holds NanoString stock or stock options. All other authors declare no competing interests.
