## [Peer Review File · Nature Communications]

Spatial transcriptomics reveals discrete tumour microenvironments and autocrine loops within ovarian cancer subclonesReviewers' Comments:

Reviewer #1:

Remarks to the Author:

In this study Denisenko et al, present the results of spacial transcriptomics combined with single cell transcriptomics examining genetic heterogeneity and the cell infiltrate in eight samples from HGSOC patients. The authors mapped the location of gene expression clusters as well as the different cell types across each section. They then use inferCNV to detect copy number alteration in the different spacial locations of the sections and successfully validated the inference by low-pass genome sequencing on microdissected regions corresponding to some of the inferred tumour subclones. The authors then examine the special expression of the four moluecular HGSOC subtypes and find that several subtypes can co-exist within sections. They suggest that tumour classification on the basis of bulk expression profiles will depend highly on the area that is being sampled, the subclones and the infiltrate of the sampled region. They then identify gene expression differences between the subclones and find genes encoding for plasma membrane and secreted proteins. The authors next examine the differences in ligand expression in malignant cells and their correlation with cognate receptors expressed on the non-malignant cells to understand how the subclones may influence infiltration of the non-malignant cells. Some of their finding included CXCL10-CXCR3 mediated recruitment of T and B cells correlating with subclones of one patient and CD47-SIRPA mediated exclusion of macrophages associated with subclones of another patient sample. Overall this is a highly interesting study and a valuable resource, it has significant implications in understanding intra and inter tumour heterogeneity in ovarian cancer and provides some insights into the cross-talk of malignant cells with the tumour microenvironment. The manuscript provides a thorough discussion of the results and the limitations of the applied technology as well as appropriate description of methods and the analysis pipeline. Sample size is admittedly modest but with a high number of Visium spots detected in each section. I have no major issue with this manuscript.

Minor comments:

Supplemental Table ST9 and ST10 have not been provided. ST10 is also not mentioned in the manuscript apart from in the list of Supplementary Tables.

Supplemental Table ST2 has not been sufficiently explained, what does the sample column refer to?

Reviewer #2:

Remarks to the Author:

The authors address an important and timely topic in investigating the spatial transcriptomic heterogeneity in high-grade serous ovarian cancer. They describe the spatial molecular profiles of eight tumor samples collected after neoadjuvant chemotherapy. Using single-cell RNAseq data and correlation based cell type proportion assessment and clonal inference verified by shallow WGS, they investigate the relationships between predicted tumor cell subclones and gene expression patterns.

The manuscript is an important effort to deconvolute and assess the spatial heterogeneity in HGSC. However there are several comments that should be addressed to strengthen the results and conclusions.

Major comments

1. Tumor cells, immune infiltration and tumor composition are assessed only via gene expression inferences. How were the spots annotated as non-malignant or malignant? Was there a cutoff for the enrichment scores? I would like to see whether the gene expression-based prediction of malignant and non-malignant cell areas would match with the histopathological assessment and visual verification.
2. Gene lists for the deconvolution score were derived from scRNAseq – can they be directly used for

Visium data or should some corrections be made in cell type enrichment analyses. This is especially when only a small fraction of the transcriptome is recovered with Visium technology, which may cause a bias in some of the lower expressing genes rendering unreliable deconvolution/cell type scores. For example, the differences between tumor cell clusters are quite small, and likely also the number of genes affect this? Has the pipeline been adjusted to work with Visium data such as in PMID 35948708? Also, How many genes from the scRNAseq data were not seen in the Visium dataset? Perhaps a venn diagram of e.g. cell-type specific genes would be good to see.

3. Tumor cell clonal inference is an interesting and relevant question. However the data needs additional clarifications and validation. For example, the inference of what was used as normal is critical when using the clonal inference from the predicted CNVs. This also is critical since the validation in the independent dataset failed to predict subclones (supplementary S24). For Supplementary Fig. 25 + methods: what were the clustering methods? Why did they exclude only the cluster 3? Cluster 1 and cluster 3 are on top of each other in the UMAP of S.Fig25d which raises the suspicion that some of the spots with normal gene expression are actually admixtures of tumor and vice versa. Also, in Patient 8 in the CNA clusters there is only one malignant cluster- what were the clustering parameters here? Again – histopathological verification and annotation of the spots is needed here to confirm the composition of the tissues to confirm malignant and non-malignant prior to the clonal inference.

4. The conclusion on the co-existence of tumor subclones with the different transcriptomic signatures is suspicious of being caused by variable tumor composition in the visium spots, as they would typically contain up to tens of single cells. Consistently, these transcriptomic subtypes have been shown to reflect rather the different cell compositions- such as immune and stromal cells, than tumor cell intrinsic gene expressions, as the authors also state themselves. For example in Figure 3, immune signature is high in all spots, mesenchymal signature also high in all spots in many of the patients. Thus, the conclusion that the profiling results of the whole tumor differs based on the area taken for analysis is not supported by the data presented, as there is no comparison to whole transcriptomics data, and across different samples and datasets. In fact, the finding presented herein is merely confirming the previous notion that the transcriptomic signatures are dependent on the tissue composition. And thus (again) a histopathological verification of the spot composition would be needed to support this finding.

5. In the transcriptomic profiling and DEG analysis of the tumor clones, it is difficult to assess from which cell types the signal is coming from. Since the CNVs are inferred from gene expression, there is likely to be a correlation between the gene expression and the clones derived from the same data. The subclone specific DEGs in the sub clone specific CNAs look like gradients which could be affected by different levels of tumor purity in the tissue spots: there seems to be a gradient in Supplementary Fig 6S which could indicate varying levels of tumor purity in the spots. Histopathological verification of the gene expressions should be verified in the tumor clones using RNA ISH with co-staining with e.g. tumor/immune cell markers.

6. The ligand -receptor analysis is interesting, and they use the scRNAseq dataset to infer the cell types and ligands. However the inference of cell infiltration and tumor cell enriched transcriptomics spots should be confirmed by histopathology annotations rather than using only the Giotto score. Additional questions:

- Were some of the tumor cell intrinsic ligands shared between patients that were differentially expressed between subclones (page 9)
- similarly, were some of the correlation patterns between non-malignant cell infiltration patterns shared between patients? (page 9)
- were the p-values corrected for multiple hypothesis testing for the correlations between ligand and cell type enrichment scores (page 9)?
- The threshold for the ligand expression and target cell enrichment is quite low in page 10 ($>0,1$, $<0,1$ in correlation analyses). I would like to see scatter plots of the data. Maybe the ligands and targets are correlated in some patients and anticorrelated in some, and this is perhaps the reason for contradictory results seen in Table 1?

7. Even though the authors seem to have selected the samples based on different chemotherapy responses, there is no information on the correlation of the responses to the spatial Transcriptomics

results.

Minor comments

1. page and row numbers are missing from the manuscript
2. CNV subclone annotations with colours is a bit confusing. I suggest annotating them with numbers at least in the text. Blue red and brown areas vs clusters is confusing – please confirm when describing clones and when areas/spots.
3. S.Figure 23 legend named “detection rate” does not match the dots in the plot
4. S.Figure 22 would be interesting but the results are quite limited? could the y-axis be clustered?
5. Figure 4c legend typo ‘patient 55’
6. Color scales in average expression scale bars are hard to interpret – different colors are used for different genes and the scale bars differ in each figure (S.Fig 21 for example)
7. No legend for S.Figure7 c) in the plot

Figures

Figure 1c - plot legend for tumor cell enrichments is missing

Figure 1d – text legend – for patients 6 and 7 – what does the colour green represent?

S.Figure 4: p-values (should be FRD corrected) are not shown in the correlation plot

S.Figure 5: scale bars should be the same within each patient to enable comparisons between the cell types

Reviewer #3:

Remarks to the Author:

Denisenko et al describe a study of ovarian cancer subclones and associated spatial microenvironment through utilizing 10x Visium (spatial) and Chromium platforms (single cell). The study is well written, and the concept/story is interesting. The figures and analyses are too descriptive, and there is a definite lack of validation. The story would benefit significantly from a more rigorous statistical examination of results, figures, as well as consideration of how to convincingly validate 1) spatial data, 2) visium inferCNV, and 3) claims about subclone specific cell neighborhoods.

Major comments:

The first three figures are largely descriptive. For figure 1, it’s not clear what cell types correspond to what colors in the Legends or the Figure. In general, I would like to see statistical tests and or rigorous analyses presented in Figures 1-3. Figures cannot be just colored plots of spatial data!

Eight samples were studied, but only 2 were focused on in Figures or described in any detail. And the 3 main figures jump between the two tumor specimens. I understand that every sample was highlighted in supp, but it was jarring to have CNV analysis on 1 sample, and receptor ligand on another.

Single-cell reference data was merged with manually curated lists for giotto cell type weight assignments into Visium data. How well does this assignment work? Can the authors compare using other assignment approaches (e.g. Seurat label transfer, Cell2loc, RCTD). It would be nice to see i) validation and ii) concordance.

The main limitation of the study is the low-resolution of the Visium technology, and potentially the limitations of inferring CNVs from this. It’s clear, looking at the Giotto annotations, and perhaps from the H&E, that the proportion of tumor cells between different regions varies significantly, this will have a large effect on CNV profiles. The authors really need to provide more confidence on this aspect, and analyses showing that subclones are not just driven by tumor cell density. I would like to see i) attempt at regressing out tumor cell density from either H&E, or giotto, and ii) better testing of

differential chromosomal regions between clones, and iii) rigorous determination of how the number of clusters was chosen for the clustering analysis for inferCNV.

For the Infer-CNV validation, which is commendable, can the authors correlate the CNV profiles or genomic bin enrichments in Figure S6 with Figure 3? While this doesn't exactly resolve the mixtures problem, as the regions for low pass CNV sequencing, it would provide confidence in the CNVs.

The ligand receptor analysis is interesting, and Figure 4 tells us there are certain receptor ligand interactions which change with immune cell proportion. But the main claim is that these changes are correlated with tumor subclone signaling.

Three main points:

- 1) The authors should present analysis in patient 5 of the subclone-CXCL10 relationship (what are the differences in CXCL10 expression between subclones, and similarly with CD47).
- 2) How do the authors disprove the alternate hypothesis: Different tissue regions have different microenvironments driven by signaling in other cell types (vs the tumor cells doing the shaping).
- 3) The authors should compare for each subclone, the relative spatial enrichment of immune cell types. The authors could use permutation testing to see if this is above a null hypothesis for differential enrichment between subclones.

We would like to thank the reviewers for their detailed comments and suggestions. These have helped to substantially improve the manuscript. We have majorly revised our work by the incorporation of exciting new single cell resolution spatial transcriptomics data using the NanoString CosMx Spatial Molecular Imager (SMI) and by substantially revising the way we identify tumour cells, infer copy number alterations and infer subclones.

All figures have either been improved (1-3) or are new (4,5). In particular we present new analyses in **Figures 4 and 5** based on CosMx single cell resolution spatial transcriptomics data.

Below we address each of the reviewers' comments in detail.

REVIEWER COMMENTS

Reviewer #1, expertise in HGSOC genomics, scRNAseq and the TME (Remarks to the Author):

In this study Denisenko et al, present the results of spatial transcriptomics combined with single cell transcriptomics examining genetic heterogeneity and the cell infiltrate in eight samples from HGSOC patients. The authors mapped the location of gene expression clusters as well as the different cell types across each section. They then use inferCNV to detect copy number alteration in the different spatial locations of the sections and successfully validated the inference by low-pass genome sequencing on microdissected regions corresponding to some of the inferred tumour subclones. The authors then examine the special expression of the four molecular HGSOC subtypes and find that several subtypes can co-exist within sections. They suggest that tumour classification on the basis of bulk expression profiles will depend highly on the area that is being sampled, the subclones and the infiltrate of the sampled region. They then identify gene expression differences between the subclones and find genes encoding for plasma membrane and secreted proteins. The authors next examine the differences in ligand expression in malignant cells and their correlation with cognate receptors expressed on the non-malignant cells to understand how the subclones may influence infiltration of the non-malignant cells. Some of their finding included CXCL10-CXCR3 mediated recruitment of T and B cells correlating with subclones of one patient and CD47-SIRPA mediated exclusion of macrophages associated with subclones of another patient sample. Overall this is a highly interesting study and a valuable resource, it has significant implications in understanding intra and inter tumour heterogeneity in ovarian cancer and provides some insights into the cross-talk of malignant cells with the tumour microenvironment. The manuscript provides a thorough discussion of the results and the limitations of the applied technology as well as appropriate description of methods and the analysis pipeline. Sample size is admittedly modest but with a high number of Visium spots detected in each section. I have no major issue with this manuscript.

We thank the reviewer for their comments. In the revision we have moved the correlation analyses between ligand expression and infiltrating populations using the Visium data to **Supplementary Note 2**. We believe the single cell resolution spatial data we generated using CosMx is much stronger than the correlation analyses we submitted in the original manuscript. Notably, we validated expression of *CXCL10* in the same tumour subclone as we identified using Visium in our original analysis. Utilising novel neighbourhood analyses using CosMx data, it was evident that the clone expressing *CXCL10* experienced higher T cell infiltration compared to the other clone. Nevertheless, the increased infiltration was not exclusive to T cells, as this clone exhibited higher overall infiltration. When assessing T cells as a proportion of all non-malignant cells and comparing these ratios between the two clones, there was no disproportionate representation.

The new neighbourhood analyses identify a *CXCL9*+ macrophage population that was preferentially associated with the *CXCL10* expressing subclone. We identify *S100A8* and *CD55* as potential ligands that drive this association. We also present evidence for subclone specific autocrine circuits.

Minor comments:

Supplemental Table ST9 and ST10 have not been provided.

Apologies for this omission. The supplementary tables have been renumbered and confirmed in this resubmission.

ST10 is also not mentioned in the manuscript apart from in the list of Supplementary Tables.

ST10 was mentioned in a **Supplementary Note**. It is now mentioned in **Supplementary Note 1** and ST11 is mentioned in **Supplementary Note 2**.

Supplemental Table ST2 has not been sufficiently explained, what does the sample column refer to?

Apologies, these are single cell library identifiers. We have now added a readme to clarify this.

Reviewer #2, expertise in ovarian cancer, TME, treatment response and scRNAseq (Remarks to the Author):

The authors address an important and timely topic in investigating the spatial transcriptomic heterogeneity in high-grade serous ovarian cancer. They describe the spatial molecular profiles of eight tumor samples collected after neoadjuvant chemotherapy. Using single-cell RNAseq data and correlation based cell type proportion assessment and clonal inference verified by shallow WGS, they investigate the relationships between predicted tumor cell subclones and gene expression patterns.

The manuscript is an important effort to deconvolute and assess the spatial heterogeneity in HGSC. However there are several comments that should be addressed to strengthen the results and conclusions.

We thank the reviewer for their comments and helpful critique of our paper. We have significantly revised our analytical workflow to address the reviewer's concerns.

Major comments

1. Tumor cells, immune infiltration and tumor composition are assessed only via gene expression inferences. How were the spots annotated as non-malignant or malignant? Was there a cutoff for the enrichment scores? I would like to see whether the gene expression-based prediction of malignant and non-malignant cell areas would match with the histopathological assessment and visual verification.

In the revised version of the manuscript we changed our approach for identification of malignant and non-malignant areas. Briefly, we used RCTD (Robust decomposition of cell type mixtures [PMID 33603203]) to identify non-malignant Visium spots (spots with tumour cell weight values below 0.15). These non-malignant spots were then used as the background in inferCNV and used to predict high-confidence CNAs in the remaining spots. The resulting inferCNV clusters were then labelled as largely malignant or largely non-malignant based on the presence of high-confidence CNAs.

To confirm that the new strategy based on RCTD reliably discriminated between malignant and non-malignant areas, we carried out a new histopathological assessment of the H&E images from the Visium sections. This confirmed that spots with high confidence CNAs and high RCTD tumour scores corresponded to regions of cells called as malignant by Qupath (**Fig 1e**).

2. Gene lists for the deconvolution score were derived from scRNAseq – can they be directly used for Visium data or should some corrections be made in cell type enrichment analyses.

We agree that robust deconvolution is critical to our inferences on cellular infiltration. We have thus used a more robust method recommended by Reviewer 3. Specifically, we have rerun our analyses using RCTD [PMID 33603203]. It has been specifically designed to use scRNA-seq

datasets to deconvolve cell type mixtures such as Visium spots. Importantly, RCTD specifically aims to tackle “platform effects” (the effects of technology-dependent library preparation on the capture rate of individual genes between sequencing platforms). This tool has been reported as one of the top performing methods in recent benchmarking studies [PMID 35577954]. Notably, as opposed to the relatively short lists of signature genes used in Giotto, RCTD uses all genes for the deconvolution. This means it is less susceptible to genes missed by the 10x 3' scRNA-seq or Visium platforms.

This is especially when only a small fraction of the transcriptome is recovered with Visium technology, which may cause a bias in some of the lower expressing genes rendering unreliable deconvolution/cell type scores. For example, the differences between tumor cell clusters are quite small, and likely also the number of genes affect this? Has the pipeline been adjusted to work with Visium data such as in PMID 35948708? Also, How many genes from the scRNAseq data were not seen in the Visium dataset? Perhaps a venn diagram of e.g. cell-type specific genes would be good to see.

Our data indicates Visium covers a high fraction of the transcriptome. In our single-cell and Visium datasets we recovered comparable numbers of genes: 18,115 genes in the single-cell dataset and 21,726 genes in the Visium dataset, with an overlap of 17,681 genes. At the level of spots and cells, Visium spots detected on average 2,740 genes whereas single cells in the scRNA-seq data detected an average of 955 genes.

3. Tumor cell clonal inference is an interesting and relevant question. However the data needs additional clarifications and validation. For example, the inference of what was used as normal is critical when using the clonal inference from the predicted CNVs. This also is critical since the validation in the independent dataset failed to predict subclones (supplementary S24).

We agree with the reviewer that clonal inference is an interesting and relevant question and that decisions in our analysis pipeline could influence the results. This is why we carried out the ultra-low pass WGS which validated the subclone-specific CNAs identified in patient 1 (**Fig. 2**).

Regarding the inference of what was normal, we have substantially revised the workflow using RCTD to robustly select background spots (see answer to question 1). The revised version of the manuscript extensively details our updated approach using RCTD and has been updated in the ‘CNA inference’ section of the methods. We now use spots with a tumour RCTD score <0.15 as the normal reference. As can be seen from **Fig.1**, clusters called with high confidence CNAs (blue, red and yellow in **Fig.1d**) overlap histologically malignant cells called by Qupath (**Fig. 1e**).

In our revised analyses, we predict areas of the slide with high confidence CNAs in all samples, however, only find evidence of subclones in samples 1, 4, 5, 6, and 7. Notably, the subclones that we predicted and validated in patient 1 are still present in the new analysis.

Regarding the failure to predict subclones in the independent dataset (from Stur *et al.*), this is somewhat expected. Their slides only had a median of 364 spots per sample in comparison to

the median of 2,507 spots per sample detected in our dataset. With almost 7-fold fewer spots per slide it will be challenging to identify subclones.

We have now included a reanalysis of 4 HGSOc samples profiled by Ferri-Borgogno *et al.* using an earlier version of the spatial transcriptomics technology (before 10x Genomics acquired it). In these larger samples we identified subclones in 2 of the four samples. Thus we have now been able to validate the presence of subclones in an independent dataset (**Supplementary Fig. S16**).

For Supplementary Fig. 25 + methods: what were the clustering methods? Why did they exclude only the cluster 3? Cluster 1 and cluster 3 are on top of each other in the UMAP of S.Fig25d which raises the suspicion that some of the spots with normal gene expression are actually admixtures of tumor and vice versa. Also, in Patient 8 in the CNA clusters there is only one malignant cluster, what were the clustering parameters here?

As discussed above we have completely revised how the background is calculated, consequently, this section on clustering has been removed and is no longer relevant. Instead, we use RCTD tumour scores below 0.15 to define spots with negligible tumour content as background.

Again – histopathological verification and annotation of the spots is needed here to confirm the composition of the tissues to confirm malignant and non-malignant prior to the clonal inference.

We thank the reviewer again for bringing up the important point about including the histopathological assessment into the manuscript. As can be seen from **Fig.1**, clusters called with high confidence CNAs (blue, red and yellow in **Fig.1d**) overlap histologically malignant cells called by Qupath (**Fig. 1e**).

4. The conclusion on the co-existence of tumor subclones with the different transcriptomic signatures is suspicious of being caused by variable tumor composition in the visium spots, as they would typically contain up to tens of single cells. Consistently, these transcriptomic subtypes have been shown to reflect rather the different cell compositions- such as immune and stromal cells, than tumor cell intrinsic gene expressions, as the authors also state themselves. For example in Figure 3, immune signature is high in all spots, mesenchymal signature also high in all spots in many of the patients. Thus, the conclusion that the profiling results of the whole tumor differs based on the area taken for analysis is not supported by the data presented, as there is no comparison to whole transcriptomics data, and across different samples and datasets. In fact, the finding presented herein is merely confirming the previous notion that the transcriptomic signatures are dependent on the tissue composition. And thus (again) a histopathological verification of the spot composition would be needed to support this finding.

We thank you for the opportunity to clarify this section. The observation that cellular composition most likely explains these transcriptional subtypes has been previously noted (in the original

submission we cited “The Impact of Stroma Admixture on Molecular Subtypes and Prognostic Gene Signatures in Serous Ovarian Cancer” PMID: 31871106).

We have now added the following text to that section to make it more obvious, that our results build upon these observations:

“These findings are in agreement with a previous study reporting that varying stroma-to-tumour cell ratios impact on the reproducibility and interpretation of these molecular subtypes³³.”

The novelty from our study is that we show that even across a very small tumour section (~5mm x 5mm) the relative signal for each of these transcriptional subtype signatures can vary such that sampling of one region of the tumour would likely classify the tumour as one subtype while sampling another region would result in a different classification.

We have reduced the text in this section and redrafted **Figure 3**. This shows that the transcriptional subtype signatures vary spatially across each section. To model comparison to whole transcriptomics data, we now include signature scores based on a pseudo bulk of all spots in each sample (shown as dashed red lines). This reconfirms that the signatures vary across each section and diverge from those from a bulk measurement.

The original title for this section “*Different subclones from the same tumour can match different molecular subtypes*” was imprecise as we did not mean to suggest that tumour subclones correspond to molecular subtypes with different tumour intrinsic transcriptional profiles. To clarify this we have now changed this to “Spatial distribution of transcriptionally defined molecular subtypes”.

5. In the transcriptomic profiling and DEG analysis of the tumor clones, it is difficult to assess from which cell types the signal is coming from. Since the CNVs are inferred from gene expression, there is likely to be a correlation between the gene expression and the clones derived from the same data. The subclone specific DEGs in the sub clone specific CNAs look like gradients which could be affected by different levels of tumor purity in the tissue spots: there seems to be a gradient in Supplementary Fig 6S which could indicate varying levels of tumor purity in the spots. Histopathological verification of the gene expressions should be verified in the tumor clones using RNA ISH with co-staining with e.g. tumor/immune cell markers.

Thank you for the comment.

In our original submission we used differential expression between inferCNV clusters in two ways.

1. The first was to confirm that clusters identified by inferCNV as having different CNVs showed significant differences in gene expression. As inferCNV uses gene expression differences; we believe this is a useful additional check.

- Secondly, to ensure that the ligands identified as putatively subclone-specific were tumour-derived, we systematically annotated every differentially expressed ligand using our scRNA-seq data and excluded those with high expression in non-cancer cells as microenvironment-derived (**Supplementary Table ST4**).

In our revision we now use the same strategy as in (**Supplementary Table ST4**) to annotate all differentially expressed genes as likely tumour-derived or microenvironment-derived.

Using NanoString CosMx Spatial Molecular Imaging (SMI), we have now tested whether genes differentially expressed between the malignant clusters P5.1 and P5.2 in patient 5 are differentially expressed between tumour subclones in the same regions. **Figure 4** confirms that the regions identified by our analyses indeed correspond to tumour cells (*CD24+*, panCK, **Fig. 4a**, **Supplementary Fig. S15**) and that the genes identified as differentially expressed by Visium are also differentially expressed between tumour cells profiled at single cell resolution (**Fig 4d**, shown below for convenience).

Notably, *IGHA1*, *IGKC* are correctly called as DE between Visium clusters, due to higher infiltration of the P5.1/*PIGR+* clone. *KRT17* and *SLPI* are significantly differentially expressed between the clones but fell below our 2 fold change threshold used for this Figure. The CosMx data also revealed that seven genes differentially expressed between P5.1 and P5.2, that we had initially categorised as stroma-derived based on scRNA-seq data (*S100A8*, *S100A9*, *CXCL9*, *INHBB*, *ADIRF*, *IGHM*, *TAGLN*), were actually highly expressed within and differentially

expressed between the *PIGR*+ and *PTGS1*+ subclones. Additionally we conclude that the Visium data underestimates the number of differentially expressed genes between these clones (due to the cell mixtures present) as the CosMx data identifies many more genes that are differentially expressed (and measurable by Visium).

6. The ligand-receptor analysis is interesting, and they use the scRNAseq dataset to infer the cell types and ligands. However the inference of cell infiltration and tumor cell enriched transcriptomics spots should be confirmed by histopathology annotations rather than using only the Giotto score.

In the original submission we used correlations between ligand expression and Giotto predicted infiltration patterns to infer relationships between ligands and cell types. In the revised version this has been moved to **Supplementary Note 2** and replaced with a more direct analysis using the CosMx data. Specifically, the CosMx data allowed us to examine tumour cell neighbourhoods and directly determine which cells and clones were expressing various ligands and receptors at single cell resolution. This has resulted in new **Figures 4** and **5**. These figures clearly show which cell types are neighbouring each tumour clone (**Fig. 4**) and the ligands and receptors each express (**Fig. 5**; note the validation of our original prediction that *CXCL10* was expressed by the P5.1/*PIGR*+ clone from patient 5).

Additional questions:

- Were some of the tumor cell intrinsic ligands shared between patients that were differentially expressed between subclones (page 9)
- similarly, were some of the correlation patterns between non-malignant cell infiltration patterns shared between patients? (page 9)
- were the p-values corrected for multiple hypothesis testing for the correlations between ligand and cell type enrichment scores (page 9)?
- The threshold for the ligand expression and target cell enrichment is quite low in page 10 ($>0,1$, $<0,1$ in correlation analyses). I would like to see scatter plots of the data. Maybe the ligands and targets are correlated in some patients and anticorrelated in some, and this is perhaps the reason for contradictory results seen in Table 1?

In the revised version of the manuscript we have removed the analyses pertaining to the above 4 questions. Instead, we now focus on using CosMx data to explore cell-to-cell communications.

We still include a correlation analysis using Visium data in the **Supplementary Note 2**, focused on correlations between ligands and RCTD scores for each cell type across all spots. We believe this is a more powerful analysis as it permits us to identify possible global correlation patterns rather than patient-specific correlations.

7. Even though the authors seem to have selected the samples based on different chemotherapy responses, there is no information on the correlation of the responses to the spatial Transcriptomics results.

As a proof of principle analysis we compared malignant spots from the three good response and three poor response samples. This revealed genes uniquely expressed by immune cells (B cells and macrophages) were more highly expressed in good response samples and genes expressed in tumour cells were more highly expressed in poor response samples. As the comparison is underpowered (3 vs 3 samples) we share these results in **Supplementary Note 1**.

Minor comments

1. page and row numbers are missing from the manuscript

Apologies, these have been added to the revised submission.

2. CNV subclone annotations with colours is a bit confusing. I suggest annotating them with numbers at least in the text. Blue red and brown areas vs clusters is confusing – please confirm when describing clones and when areas/spots.

Thank you for this suggestion, we now refer to the clusters by their numbers.

Also, thank you very much for pointing out that we used the terms areas/clusters/subclones somewhat loosely. We have endeavoured to fix this in the resubmission. We refer to clusters, malignant clusters as those with high confidence CNAs and only use the term subclone to refer to tumour cells.

3. S.Figure 23 legend named “detection rate” does not match the dots in the plot

Thank you for noticing this. We made sure legends are correct in all our supplementary dot plots.

4. S.Figure 22 would be interesting but the results are quite limited? could the y-axis be clustered?

Thank you for this suggestion. We have substantially changed this figure (Supplementary Fig. S14 in the resubmission), but we have also added clustering of the y-axis.

5. Figure 4c legend typo ‘patient 55’

Thank you, this has been removed from the manuscript.

6. Color scales in average expression scale bars are hard to interpret – different colors are used for different genes and the scale bars differ in each figure (S.Fig 21 for example)

Thank you for this suggestion, the corresponding dotplots have been removed from the revised manuscript.

7. No legend for S.Figure7 c) in the plot

We have updated the legend.

c) Projection of the CNA-based clusters onto the tissue section as Visium spots. Blue and red spots correspond to putative tumour subclones, grey spots are non-malignant regions with RCTD tumour scores <0.15 , green and pink correspond to border regions.”

Figures

Figure 1c - plot legend for tumor cell enrichments is missing

Thank you, this is updated in the resubmission.

Figure 1d – text legend – for patients 6 and 7 – what does the colour green represent?

Thank you, this is updated in the resubmission.

S.Figure 4: p-values (should be FRD corrected) are not shown in the correlation plot

Thank you for suggesting this (the corresponding Figure was renumbered as **Supplementary Fig. S2** in the revised manuscript). We have now calculated FDR-corrected p-values. All but 3 correlation coefficients shown on this plot are significant. To avoid overplotting, we include this information in the legend.

S.Figure 5: scale bars should be the same within each patient to enable comparisons between the cell types

Thank you for highlighting this. This figure (**Supplementary Fig. S3** in the revised manuscript) now shows RCTD cell type weights which are on the same scale.

Reviewer #3, expertise in spatial transcriptomics and scRNAseq (Remarks to the Author):

Denisenko et al describe a study of ovarian cancer subclones and associated spatial microenvironment through utilizing 10x Visium (spatial) and Chromium platforms (single cell). The study is well written, and the concept/story is interesting. The figures and analyses are too descriptive, and there is a definite lack of validation. The story would benefit significantly from a more rigorous statistical examination of results, figures, as well as consideration of how to convincingly validate 1) spatial data, 2) visium inferCNV, and 3) claims about subclone specific cell neighborhoods.

We thank the reviewer for their comments. We have applied additional more rigorous tests to the data to strengthen our claims of subclones. We have also now generated new single molecule imaging data using the NanoString CosMx platform to validate our spatial data and perform neighbourhood analyses at single-cell resolution.

Major comments:

The first three figures are largely descriptive. For figure 1, it's not clear what cell types correspond to what colors in the Legends or the Figure. In general, I would like to see statistical tests and or rigorous analyses presented in Figures 1-3. Figures cannot be just colored plots of spatial data!

The primary aim of **Figure 1** was to give a graphical overview of the data. We have updated this figure with the following changes:

1. We now use RCTD to show tumour cell enrichment in **Fig.1c**. We replaced the Giotto analysis as RCTD generates global scores which can be compared across samples. It has also been reported as one of the top performing methods in recent benchmarking studies [PMID: 35577954].
2. We have added a new row to the figure showing histopathology (tumour cell annotations) based on expert annotation using the QuPath tool on the H&E images (**Fig. 1e**).
3. We have generated a Supplementary Sankey diagram that summarises the relationship between each spot level annotation shown in **Figure 1**.
4. We have included Adjusted Rand Index and Normalised Mutual Information to measure the similarity of spot level annotations shown in **Figure 1**.
5. We have updated the colours and figure legend to improve readability (see below). Yellow-highlighted text indicates changes.

Figure 1. Graphical overview of the Visium data generated for eight HGSOC samples. a) Hematoxylin (blue) and eosin (red) stained tissue sections. Patient IDs are shown for each section. **b)** Gene expression-based clustering of the Visium data. Expression profiles for spots are clustered and then mapped back onto the tissue

sections. Cluster colours are randomly assigned. **c)** Tumour cell enrichment weights calculated using RCTD²⁷. Spots with tumour cell enrichment are shown in red. **d)** CNA-based clusters. Blue, red, and yellow spots correspond to putative tumour subclones, grey spots are non-malignant regions with RCTD tumour scores <0.15, green and pink correspond to border regions. **e)** Histopathological expert annotation of the tissue sections using QuPath⁷⁵. Red corresponds to malignant cells, green corresponds to stroma. Note that the colours shown in (b) are arbitrary but highlight that unsupervised clustering of the expression data using Seurat (b), and clustering of inferCNV profiles (d) identified clusters with spatial patterns that largely reflected the morphology shown in (a) and the pathology shown in (e). **Supplementary Fig. S19** shows Sankey diagrams and statistics summarising the relationship between the clusters shown in (b) and (d).

We have also updated **Figure 2** and the related part of the manuscript with the following changes:

1. Using Hidden Markov and Bayesian latent mixture modelling within inferCNV we now only focus on high-confidence CNAs. This added statistical rigour to our analyses. These are shown in **Figure 2c** and **Supplementary Figures S5b - S11b**.
2. We calculated the correlation between inferCNV signal (**2b**) and IchorCNV results (**2e**) and included the following in the manuscript:

“Notably, the ichorCNV values were strongly correlated with the averaged inferCNV signal across spots within the respective cluster, with Spearman correlation coefficients of 0.67, 0.68, and 0.71 for the clusters P1.1, P1.2, and P1.3, respectively.”

3. We replaced cluster labelling from colour-coding to using cluster IDs to improve readability.

We have updated **Figure 3** with the following changes:

1. We performed significance testing using Mann-Whitney U test.
2. We added red dotted lines showing the subtype scores calculated using a pseudo-bulk of all spots.
3. We added the results for patient 1 to address the next comment.
4. We expanded the legend for readability.

The updated legend is as follows:

Figure 3. Variations in HGSOC molecular subtype signatures. **a)** Distribution of Module scores for the four molecular subtype signatures in each of the CNA-based clusters in patient 5. Black lines are medians. Red dotted lines show the value if all spots are combined as a pseudo-bulk. “ns” indicates non-significant differences; all other pairwise comparisons returned significant results, significance was determined using

Mann-Whitney U test. **b)** Spatial distribution of Module scores for the four molecular subtype signatures in patient 5. **c)** Distribution of Module scores for the four molecular subtype signatures in each of the CNA-based clusters in patient 1. Black lines are medians. Red dotted lines show the value if all spots are combined as a pseudo-bulk. “ns” indicates non-significant differences; all other pairwise comparisons returned significant results, significance was determined using Mann-Whitney U test with Benjamini-Hochberg correction and 0.05 threshold. **d)** Spatial distribution of Module scores for the four molecular subtype signatures in patient 1. The labels shown correspond to the mesenchymal (C1.MES), immunoreactive (C2.IMM), differentiated (C4.DIF), and proliferative (C5.PRO) subtypes respectively reported in²⁵.

Eight samples were studied, but only 2 were focused on in Figures or described in any detail. And the 3 main figures jump between the two tumor specimens. I understand that every sample was highlighted in supp, but it was jarring to have CNV analysis on 1 sample, and receptor ligand on another.

We have kept the focus on samples 1 and 5 in **Figures 2-5** but have attempted to improve the flow. We start with **Figure 1** which showcases all of the samples studied and summarises morphology, gene expression clusters, inferred tumour cell content, inferred CNA clusters and histopathology. The **Supplementary Figures** provide a more detailed view for each sample.

We then show that the predicted CNAs are real, by validating them with microdissection and WGS. We deliberately focus on patient 1 who had the highest number of predicted subclones (**Figure 2**).

To improve the flow we include both patient 1 and patient 5 samples in **Figure 3** to show variation of transcriptional subtype signatures across sections.

We then focus on patient 5 in **Figures 4 & 5** using CosMx to validate subclone-specific neighbourhoods and subclone-specific differential expression of ligands and receptors.

Single-cell reference data was merged with manually curated lists for giotto cell type weight assignments into Visium data. How well does this assignment work? Can the authors compare using other assignment approaches (e.g. Seurat label transfer, Cell2loc, RCTD). It would be nice to see i) validation and ii) concordance.

Thank you for the suggestion of other assignment approaches. In the revised version of the manuscript, we decided to use RCTD [PMID 33603203] instead of Giotto, since this tool has been reported as one of the top performing methods in recent benchmarking studies [PMID: 35577954]. We chose this as it uses the entire single cell RNA-seq matrix instead of using selected markers, and that the RCTD scores are comparable across each sample.

As benchmarking of these tools has already recently been carried out by others, we do not include a comparison with other assignment approaches. We removed the Giotto analyses from the manuscript completely, however, the Giotto scores correlated well with RCTD for most cell types - Spearman correlation coefficients of 0.86 for Fibro 5, 0.78 for B/Plasma cells, 0.77 for Tumour cells, 0.66 for Macrophages, 0.49 for Myofibroblasts, 0.36 for T cells, and 0.19 for Mesothelial cells.

The main limitation of the study is the low-resolution of the Visium technology, and potentially the limitations of inferring CNVs from this. It's clear, looking at the Giotto annotations, and perhaps from the H&E, that the proportion of tumor cells between different regions varies significantly, this will have a large effect on CNV profiles. The authors really need to provide more confidence on this aspect, and analyses showing that subclones are not just driven by tumor cell density.

We agree with the reviewer that the low resolution of the Visium technology is a limitation of our study and we understand the reviewer's concern regarding inferring CNV from Visium data. We would like to highlight, however, that this approach (inferCNV on Visium) has already been used by others, and the corresponding study was recently published in *Nature* (PMID 35948708). Importantly, that study specifically explored the applicability of inferCNV to Visium data by performing an *in silico* simulation and confirmed that "inferCNV could robustly capture sufficient and accurate CNV information for individual spots from a multifocal tumour model". This, together with the fact that our ultra-low-pass DNA sequencing confirmed the predicted CNVs in patient 1, and that our new CosMx data validated differential expression of tumour-expressed genes between subclones in patient 5, we believe, supports the validity of our approach.

I would like to see i) attempt at regressing out tumor cell density from either H&E, or giotto, and ii) better testing of differential chromosomal regions between clones, and iii) rigorous determination of how the number of clusters was chosen for the clustering analysis for inferCNV.

Underlying these suggestions is the possibility that some of the clusters identified by inferCNV could just be due to differences in tumour proportion within each spot. In response, logically, if there were no different subclones, but just one clone present at different proportions, we would expect to see the strongest CNA evidence in the inferCNV cluster with the highest proportion of tumour cells while areas with a lower proportion of tumour cells would likely have the same (or a subset) of these CNAs with weaker signal. We see evidence for this in some clusters that we label as border regions. However, for the manuscript we focus on clusters where there are clear subclone specific CNAs (e.g., subclone 1 has a deletion on Chr1 not observed in subclone 2, and subclone 2 has an amplification on Chr8 not observed in subclone 1).

In our revised version of the manuscript we are more conservative in our estimates on numbers of subclones. Firstly, we have used RCTD to identify regions as likely tumour or non-tumour.

Non-tumour spots (predicted tumour cell content of less than 0.15) are used as background and CNA signal is inferred for the remaining spots with predicted tumour cell content above 0.15. Spots are clustered based on their inferred CNA profiles and we then apply Hidden Markov and Bayesian latent mixture modelling in inferCNV to predict high-confidence CNAs in the resulting clusters. Clusters with few (or no) high-confidence CNAs are annotated as likely non-malignant. Clusters with high-confidence CNAs are then compared. Neighbouring clusters are then examined to determine whether they could correspond to mixtures. If a neighbouring cluster has no unique high confidence CNAs and only contains CNAs called in another cluster we label it as likely border.

To address the reviewer's comments we detail evidence for subclones in 5 of the 8 patient samples.

In Patient 1 we observe 5 clusters and predict **three** high confidence CNA clonotypes:

- P1.1 - Multiple high confidence CNAs, unique deletion in Chr5*
- P1.2 - Multiple high confidence CNAs, unique amplification of Chr1 and deletion in Chr4*
- P1.3 - Multiple high confidence CNAs, unique amplifications on Chr3, Chr5, Chr11 and deletion on Chr8*
- P1.4 - likely border (all high confidence CNAs are found in other clusters)
- P1.5 - likely border (all high confidence CNAs are found in other clusters)

***Note the CNAs in P1.1, P1.2, P1.3 were validated by low pass WGS in Fig. 2.**

In Patient 2 we observe 3 clusters and predict **one** high confidence CNA clonotype:

- P2.1 Multiple high confidence CNAs
- P2.3 Likely border (all high confidence CNAs are shared with P2.1)
- P2.2 no high confidence CNAs

In Patient 3 we observe 3 clusters and predict **one** high confidence CNA clonotype:

- P3.1 Multiple high confidence CNAs, multiple CNAs not observed in P3.3
- P3.2 Likely border. One small high confidence deletion on Chr14.
- P3.3 Likely border. One small high confidence amplification on Chr14. All other high-confidence CNAs are seen in P3.1.

In Patient 4 we observe 4 clusters and predict **two** high confidence CNA clonotypes:

- P4.1 Multiple high confidence CNAs including a unique amplification on Chr X.
- P4.2 Multiple high confidence CNAs including a unique CNA on Chr 14 and stronger amplification on Chr 1.
- P4.3 Likely border. Three high confidence CNAs shared with P4.1 and P4.2. One unique high confidence CNA predicted on Chr8 but seen in P4.1 (but not called as significant).
- P4.4 Likely border. Shares ChrX amplification from P4.1

In Patient 5 we observe 3 clusters and predict **two** high confidence CNA clonotypes:

- P5.1 Unique high confidence CNAs on Chr11,12*
- P5.2 Multiple unique high confidence CNAs (e.g., on Chr1,2,3,4,5,7,17)*
- P5.3 Likely border. One small high confidence deletion on Chr20 seen in P5.1 and P5.2.

***Note, differential expression between these two clones is validated by the new CosMx data.**

In Patient 6 we observe 4 clusters and predict **two** high confidence CNA clonotypes:

- P6.1 Multiple unique high confidence CNAs (e.g., on 8, 11, 14, 16, 17, X)
- P6.4 Likely border containing clonotype from P6.1. No unique high confidence CNAs.
- P6.2 Multiple unique high confidence CNAs (e.g. on Chr1, 2 and 7)
- P6.3 Likely border containing clonotype from P6.2. No unique high confidence CNAs.

In Patient 7 we observe 3 clusters and predict **two** high confidence CNA clonotypes:

- P7.1 Multiple unique high confidence CNAs (e.g., on chr 5,6,13,14,18)
- P7.3 no high confidence CNAs
- P7.2 Multiple unique high confidence CNAs (e.g., on chr 4,11,19)

In Patient 8 we observe 2 clusters and predict **one** high confidence CNA clonotype:

- P8.1 multiple high confidence CNAs
- P8.2 no high confidence CNAs.

With this conservative approach we reduce the frequency of observed CNAs to one sample with 3 high confidence subclones, four samples with 2 high confidence subclones and three samples with only one high confidence clonotype.

For the Infer-CNV validation, which is commendable, can the authors correlate the CNV profiles or genomic bin enrichments in Figure S6 with Figure 3? While this doesn't exactly resolve the mixtures problem, as the regions for low pass CNV sequencing, it would provide confidence in the CNVs.

We thank the reviewer for this important suggestion. In the revised manuscript we include the following:

“Notably, the ichorCNV values were strongly correlated with the averaged inferCNV signal across spots within the respective cluster, with Spearman correlation coefficients of 0.67, 0.68, and 0.71 for the clusters P1.1, P1.2, and P1.3, respectively.”

The ligand receptor analysis is interesting, and Figure 4 tells us there are certain receptor ligand interactions which change with immune cell proportion. But the main claim is that these changes are correlated with tumor subclone signaling.

In the revised version of the manuscript we completely changed the approach for cell-to-cell communication analysis. In the newly generated **Figures 4 and 5**, we now focus on the CosMx dataset for the neighbourhood and cell-to-cell communication analysis, since it provides single-cell resolution. We still include the Visium-based correlation analysis as a **Supplementary Note** (and for this we changed our approach from subclone-oriented and within-patient to global correlation).

Three main points:

1) The authors should present analysis in patient 5 of the subclone-CXCL10 relationship (what are the differences in CXCL10 expression between subclones, and similarly with CD47).

As mentioned above we have switched to using the CosMx data to study subclone specific signalling in patient 5 (**Figures 4 and 5**). Notably, as predicted by our original analyses, *CXCL10* is differentially expressed between the subclones both according to the Visium data (\log_2FC 0.9, FDR 1.9×10^{-4}) and CosMx data (\log_2FC 1.6, FDR 10^{-22}). We also confirm *CXCL10-CXCR3* signalling between the *PIGR+* subclone and neighbouring *CCL5+* T cells is plausible (**Figure 5c**).

2) How do the authors disprove the alternate hypothesis: Different tissue regions have different microenvironments driven by signaling in other cell types (vs the tumor cells doing the shaping).

We agree with the reviewer that other cell types likely contribute to 'the shaping'. Our analyses report differences in cellular makeup of tumour areas with different subclones. In our new neighbourhood analyses based on the CosMx data for patient 5 we observed that both clones were neighboured by very similar mixes of cell types (**Fig. 5a**). We then focused on those that were overrepresented near each clone. Unlike in our previous analyses based on correlations, there was not an over-representation of T cells near one clone vs the other, however, we did observe over-representation of *CXCL9*+ macrophages near the *PIGR*+ clone and of *COL3A1*+ fibroblasts and *ENG*+ endothelial cells near the *PTGS1*+ clone (**Fig 5b**).

3) The authors should compare for each subclone, the relative spatial enrichment of immune cell types. The authors could use permutation testing to see if this is above a null hypothesis for differential enrichment between subclones.

Thank you for this suggestion. As mentioned above, the enrichment was re-calculated using RCTD instead of Giotto. As suggested by the reviewer we have additionally performed the permutation testing, and included this as **Supplementary Fig. S14**.

Reviewers' Comments:

Reviewer #1:

Remarks to the Author:

The authors have addressed my concerns, I support this manuscript for publication.

Reviewer #2:

Remarks to the Author:

Denisenko et al have revised the manuscript, and added CosMX analysis on one tumor sample, scRNAseq and ligand receptor analyses. The added data indicate the presence of transcriptionally distinct tumor cell subpopulations in ovarian cancer, which are interesting and convincingly shown in the manuscript.

1. My main concern is that even though the authors have the pathologist annotations for the tumor cell enriched spots, they are not using them in the analyses. I would like the authors to reproduce the analyses for clonal inference using only the pathologist annotated tumor enriched spots. RCDT values for tumor are somewhat inconsistent with the pathology annotations (Figure 2). This puts in question the main conclusions of the paper showing InferCNV predicted, supposedly tumor genetic CNA subclones in areas where the pathologist has not annotated tumor. This supports the alternative hypothesis that the inferCNV "subclones" are caused by just variations in tumor purity. This is very evident in e.g. patient 1 where RCTD based tumor estimation is allowing for the CNA based clusters to be called in the upper are of the sample ("sub clone" P1.2), where no tumor cells are found by the pathologist. Similarly this is true for patients 3, 4, 6, 7, 8. In fact, that area in Fig 2 D looks necrotic thus confounding the analyses.

2. The interpretation of sWGS validation results is based only on visual inspection of the graphs. Rather the sWGS CNAs should be quantitated and investigated more properly to really show tumor CNA subclones. Looking at the CNA plots of the replicates in Figure S13, markedly differing profiles can be observed even between the three replicates for "clone" P1.3. and other "clones" as well. Is the CNA profile difference between "clones" quantitatively significantly bigger than the inter-replicate heterogeneity? The estimated tumor fraction is also significantly lower than in P1.1., and it is suspicious that the predicted ploidy is around 2 in "normal" samples. The sWGS plots in figure 2E, do raise the question whether the different CNA profiles are variations of tumor purities/technical, and thus the sWGS interpretation does not here validate the transcriptomics inferred "clones". Moreover, the genomic validation should be performed on more than just one sample.

3. The inherent problem which hasn't been addressed is also the possible bias caused by stromal/immune cell infiltration. This factor unfortunately cannot be validated by histopathological annotation of tumor and non-malignant regions, and its importance is shown e.g. in Fig. 3 where C2.IMM signature is scoring quite high in both presented samples. In addition, the DEG comparison of the cancer subclones, which is presented in the section "Gene expression and microenvironment differences between 28 tumour subclones", reveals substantial number of immune-derived genes as factors differentiating between clones. That highlights the importance of immune cells in shaping the transcriptomic profiles of Visium spots. Thus it cannot be reasoned that the tumor cells affect the immune microenvironment, but rather just that there are spatially different tumor-immune signatures in the ovarian cancer TME. With the presented data, the interpretation on the different cancer subclones eliciting different immune microenvironments cannot be made. Thus the conclusions on cancer genetic clones should be toned down (including in the title) and focused on transcriptionally distinct tumor cell populations.

Reviewer #3:

Remarks to the Author:

The reviewers have addressed my concerns regarding the Visium data. Given the new data on Cosmx, I had the following concerns:

In general, I would like to see more validation and QC data for the Cosmx.

Many FOVs look very sparse in Cosmx data in 4D. What fraction of cells were assigned to a cell in Cosmx? Are there cells that are left out in certain FOVs?

There seems to be some mixing in marker gene plots. The immune populations look especially mixed in 4A and 4B, in the UMAP and marker plots. Can the authors comment on this?

It is possible that the authors may want to use a higher level macrophage calling, as those seem particularly mixed, this will have a large effect on neighborhood analyses in Figure 5. I would suggest seeing the sensitivity of those analyses to granularity of cell type calling. Can the authors use single-cell data to try to deconvolve this?

What was the number of transcripts/cell for the Cosmx data?

Can correlation analysis be done between spots and images in Cosmx vs Visium?

REVIEWER COMMENTS

Reviewer #1 (Remarks to the Author):

The authors have addressed my concerns, I support this manuscript for publication.

Reviewer #2 (Remarks to the Author):

Denisenko et al have revised the manuscript, and added CosMX analysis on one tumor sample, scRNAseq and ligand receptor analyses. The added data indicate the presence of transcriptionally distinct tumor cell subpopulations in ovarian cancer, which are interesting and convincingly shown in the manuscript.

We are glad that the reviewer is convinced by the presence of transcriptionally distinct tumour cell subpopulations. The remaining point is whether these represent subclones with different copy number alterations. In the previous revision we used a conservative approach to annotate potential subclones. Our requirement was that to identify a pair of subclones, each needed to possess high confidence CNAs that were not observed in the other putative subclone. Such mutually exclusive CNAs can not be explained by tumour:stroma ratios.

Logically, when only one clone is present at varying proportions, we anticipate the Visium spots with the highest proportion of tumour cells to have the most robust Copy Number Alteration (CNA) predicted by inferCNV. In spots with more stromal cells and a lower proportion of tumour cells a weaker signal for the same genomic regions is expected and in some cases a CNA may be missed altogether. **It's important to note however that an increase in the amount of stromal cells should never result in the observation of additional unique CNAs.**

In the examples below we highlight (with coloured arrows) mutually exclusive CNAs that can not be explained by tumour:stroma ratios. For all subclones described in the manuscript we require the existence of mutually exclusive CNAs.

In the case of Patient 1 we observed 5 clusters and predicted **three** high confidence CNA clonotypes:

- P1.1 - Multiple high confidence CNAs, unique deletion in Chr5*
- P1.2 - Multiple high confidence CNAs, unique amplification of Chr1 and deletion in Chr4*
- P1.3 - Multiple high confidence CNAs, unique amplifications on Chr3, Chr5, Chr11 and deletion on Chr8*
- P1.4 - likely border (all high confidence CNAs are found in P1.1)
- P1.5 - likely border (all high confidence CNAs are found in P1.1)

Similarly mutually exclusive unique CNAs were observed in an independent spatial transcriptomics dataset (**Supplementary figure 16c**).

Taken together with the WGS on microdissected tissue and CosMx validations we believe we have gone far enough to demonstrate the presence of tumour subclones with distinct CNAs.

1. My main concern is that even though the authors have the pathologist annotations for the tumor cell enriched spots, they are not using them in the analyses. I would like the authors to reproduce the analyses for clonal inference using only the pathologist annotated tumor enriched spots. RCDT values for tumor are somewhat inconsistent with the pathology annotations (Figure 2). This puts in question the main conclusions of the paper showing InferCNV predicted, supposedly tumor genetic CNA subclones in areas where the pathologist has not annotated tumor. This supports the alternative hypothesis that the inferCNV “subclones” are caused by just variations in tumor purity. This is very evident in e.g. patient 1 where RCDT based tumor estimation is allowing for the CNA based clusters to be called in the upper area of the sample (“sub clone” P1.2), where no tumor cells are found by the pathologist. Similarly this is true for patients 3, 4, 6, 7, 8. In fact, that area in Fig 2 D looks necrotic thus confounding the analyses.

We thank the reviewer for the comments. We now complement the original Qupath images we included in **Figure 1e** with closeup views of the H&E and tumour cells identified in the regions containing each subclone (in **Fig.2f**, **Supp Figs S7-S10**). In all cases, tumour cells are present and called by Qupath. Hyperchromatic, pleomorphic nuclei, arranged in variably solid, glandular and papillary patterns typical of HGSOC are visible. We trust this better demonstrates that the regions where we call high confidence sub-clones contain tumour cells. In the example that the reviewer refers to - the upper area in patient 1 (subclone P1.2) - there are indeed tumour cells called by Qupath (**Fig.2f - centre**). These are more loosely packed than in the other regions and have large nuclei but are still called tumour cells. There is no evidence of necrosis in this region.

a

* boxes indicate areas shown above

As mentioned in the previous section, the unique CNAs observed in clones P1.1, P1.2 and P1.3 can not be explained by differences in tumour purity.

We would also like to highlight that this approach (inferCNV on Visium) has already been used in Joakim Lundeberg's paper in Nature on spatially inferred CNAs. <https://www.nature.com/articles/s41586-022-05023-2>. Importantly, that study specifically explored the applicability of inferCNV to Visium data by performing an *in silico* simulation and confirmed that "inferCNV could robustly capture sufficient and accurate CNV information for individual spots from a multifocal tumour model".

Note our requirement for **mutually exclusive high confidence CNAs is conservative** and goes one step further in terms of evidence than that of the Lundeberg publication in Nature.

2. The interpretation of sWGS validation results is based only on visual inspection of the graphs. Rather the sWGS CNAs should be quantitated and investigated more properly to really show tumor CNA subclones. Looking at the CNA plots of the replicates in Figure S13, markedly differing profiles can be observed even between the three replicates for "clone" P1.3. and other "clones" as well. Is the CNA profile difference between "clones" quantitatively significantly bigger than the inter-replicate heterogeneity? The estimated tumor fraction is also significantly lower than in P1.1., and it is suspicious that the predicted ploidy is around 2 in "normal" samples. The sWGS plots in figure 2E, do raise the question whether the different CNA profiles are variations of tumor purities/technical, and thus the sWGS interpretation does not here validate the transcriptomics inferred "clones".

In our previous revision we included the following:

"Notably, the ichorCNV values were strongly correlated with the averaged inferCNV signal across spots within the respective cluster, with Spearman correlation coefficients of 0.66, 0.67, and 0.65 for the clusters P1.1, P1.2, and P1.3, respectively. Moreover, the WGS confirmed the predicted high-confidence CNAs, e.g. large amplifications at chromosomes 8, 12, and 20 and deletions at chromosomes 6, 17, and 19 predicted in all malignant clusters (Fig. 2e). It also validated multiple predicted cluster-specific high-confidence CNAs, including a deletion in chromosome 5 in cluster P1.1, an amplification in chromosome 1 and a deletion in chromosome 4 in cluster P1.2, and an amplification in chromosome 19 in cluster P1.3 (Fig. 2c,e). "

We have now replaced **Figure 2e** to specifically highlight mutually exclusive (and shared) CNAs validated across each of the replicates from **Figure S13**. We believe this better highlights the mutually exclusive CNAs that were predicted by inferCNV and validated by low pass WGS.

We also updated the text to reflect the changes:

“Furthermore, the WGS confirmed multiple high-confidence CNAs predicted by inferCNV, including those specific to the three clusters and others shared among all malignant clusters. Specifically, the deletion at chromosome 4 in P1.2 was validated across all three replicates, the amplification at chromosome 4 in P1.3 was validated in both replicates, the deletion at chromosome 5 in P1.1 was validated in both replicates, and the amplification at chromosome 19 in P1.3 was confirmed in both replicates. Additionally, the amplification of chromosome 12, which was common to all malignant clusters, was also validated (**Fig. 2e**). **Supplementary Fig. S13** shows genome-wide views of the ichorCNV results. Note the observed patterns of validated cluster specific CNAs cannot be explained by differences in tumour-stroma proportions, thus, we concluded that tissue areas corresponding to the clusters P1.1, P1.2, and P1.3 likely contain tumour subclones that are closely related, sharing several CNAs, but also possessing additional unique CNAs.”

Moreover, the genomic validation should be performed on more than just one sample.

Additional genomic validation will not substantially change the conclusions of the paper. Our claims on the presence of subclones is supported by:

- Observation of high confidence mutually exclusive CNAs across multiple patients in both our data and that of Ferri-Borgogno *et al.*,
- Lowpass WGS validation of subclone specific CNAs in patient 1
- CosMx validation of different transcriptional profiles of subclones inferred in patient 5

Note that we go further than Joakim Lundeberg's paper in Nature on spatially inferred CNAs <https://www.nature.com/articles/s41586-022-05023-2>.

- We require mutually exclusive high confidence CNAs to call an inferCNV cluster a clone.
- We validate multiple CNAs by using microdissected WGS (whereas they used DNA fish against 1 loci *EHD2* to validate siCNAs in one sample)

3.The inherent problem which hasn't been addressed is also the possible bias caused by stromal/immune cell infiltration. This factor unfortunately cannot be validated by histopathological annotation of tumor and non-malignant regions, and its importance is shown e.g. in Fig. 3 where C2.IMM signature is scoring quite high in both presented samples. In addition, the DEG comparison of the cancer subclones, which is presented in the section "Gene expression and microenvironment differences between tumour subclones", reveals substantial number of immune-derived genes as factors differentiating between clones. That highlights the importance of immune cells in shaping the transcriptomic profiles of Visium spots. Thus it cannot be reasoned that the tumor cells affect the immune microenvironment, but rather just that there are spatially different tumor-immune signatures in the ovarian cancer TME. With the presented data, the interpretation on the different cancer subclones eliciting different immune microenvironments cannot be made. Thus the conclusions on cancer genetic clones should be toned down (including in the title) and focused on transcriptionally distinct tumor cell populations.

As we presented in response to the previous points we have already provided strong evidence that genetic subclones with mutually exclusive CNAs are observed in our data and that of Ferri-Borgogno *et al.*

In terms of the language we have considered this carefully. The CosMx data clearly shows that in patient 5 i) ligands and receptors are differentially expressed between subclones ii) there are statistically significant differences in the proportions of different non-tumour cell types neighbouring the *PIGR+* and *PTGS1+* clones. From this we are confident to state that the tumour cells interactions are **likely** to be modulated (because the subclones express different ligands) and that tumour subclones **may** influence infiltrations patterns (because we directly see statistically significant differences in the infiltration of non-tumour cells in the CosMx data and predict differences for the other patients based on the RCTD decomposition of the Visium data).

We have reviewed all statements on these points throughout the manuscript and have made the following changes.

1. "Title: Spatial transcriptomics reveals discrete tumour microenvironments and autocrine loops within ovarian cancer subclones"

>>'reveals' to 'predicts'

2. Abstract: "We hypothesise that this may modulate their interactions with stromal and immune cells."

>> no change

3. Abstract: "...and that subclone-specific ligand and receptor expression patterns likely modulate how these tumour cells interact with their local microenvironment.

>> no change

4. Intro: "that tumour cells may influence their local microenvironment by subclone-specific upregulation of ligands and receptors"

>> no change

5. Intro: "Our analyses provide a likely link between subclonal genotype differences and differential infiltration patterns."

>> change 'provide a likely link' to 'predict a link'

6. Discussion: "This raises the possibility that tumour subclones may shape the composition of their local microenvironments."

>>no change

7. Discussion: "...subclones expressing different ligands may modulate their local tumour microenvironment"

>> no change

Reviewer #3 (Remarks to the Author):

The reviewers have addressed my concerns regarding the Visium data. Given the new data on Cosmx, I had the following concerns:

In general, I would like to see more validation and QC data for the Cosmx.

1. Many FOVs look very sparse in Cosmx data in 4D. What fraction of cells were assigned to a cell in Cosmx? Are there cells that are left out in certain FOVs?

Based on the cellpose segmentation results, we initially obtained 39,939 cell segments. After excluding segments with less than 100 transcripts, 26,895 putative cells were kept for downstream analysis. This corresponds to 67% of initial cell segments.

The number of putative cells (cells with ≥ 100 transcripts) detected per FOV ranged from 156 (FOV2) to 2986 (FOV22). As the reviewer notes, the segments towards the middle of the section had lower numbers of putative cells detected compared to those containing the *PIGR+* and *PTGS1+* subclones. Below is a visual summary of the cell segments detected by cellpose and the number remaining after thresholding, which we now added to **Supplementary Fig. S15**.

Number of cell segments identified by Cellpose in each Field Of View (FOV) after/before filtering (≥ 100 transcripts).

2. There seems to be some mixing in marker gene plots. The immune populations look especially mixed in 4A and 4B, in the UMAP and marker plots. Can the authors comment on this?

We believe this could be due to two possible factors:

1. The 960 gene panel used was not optimised for separation of cell types in HGSOC tissue. This issue could possibly be addressed by use of a custom panel targeting marker genes of cell types in HGSOC however this is beyond the scope of this study.
2. Possible 'segmentation doublets' where the cellpose segmentation has failed to split adjacent cells and they are merged into one cell segment.

To address the possibility of segmentation doublets we used Scrublet (<https://doi.org/10.1016/j.cels.2018.11.005>) to identify likely doublets amongst the 26,895 putative cells with ≥ 100 transcripts. After doublet filtering 21,651 cells remained (19.5% of segments were discarded as potential segmentation doublets). We have updated the text to reflect this and rerun all clustering, annotation, differential expression, neighbourhood and cell-to-cell communication analyses of the filtered data.

3. It is possible that the authors may want to use a higher level macrophage calling, as those seem particularly mixed, this will have a large effect on neighborhood analyses in Figure 5. I would suggest seeing the sensitivity of those analyses to granularity of cell type calling. Can the authors use single-cell data to try to deconvolve this?

Repeating the Leiden clustering on the cells identified as singlets by Scrublet identified 19 clusters which were then collapsed into 12 putative cell types based on the presence or absence of cluster specific markers (of the 19 clusters 2, 3, 3, 2, and 2 were merged into *LYZ*⁺ macrophages, *COL3A1*⁺ fibroblasts, *CCL5*⁺ T cells, *SPP1*⁺ macrophages and *PTGS1*⁺ tumour cells respectively) (**Fig. 4a,b**). This is almost the same cell types as reported in the last revision however we collapsed the previously identified *MMP9*⁺ and *SPP1*⁺ macrophages into one population (the *MPP9*⁺ cells appear to be a subpopulation of *SPP1*⁺ cells; Note: neither are enriched near tumour cells) and the minor *TWIST1*⁺ population was lost.

Of the remaining macrophage populations, the *LYZ*⁺, *C1QC*⁺ and *CXCL9*⁺ populations exhibited sufficient differences in marker gene expression (**Fig 4b**) and relative enrichment/depletion near tumour cells and subclones (**Fig 5ab**) to conclude that these cells are distinct macrophage populations.

Note, improving on the previous neighbourhood analyses, we implemented significance testing based on cell label permutation to (i) test whether particular cell types are over or under represented near tumour cells and (ii) whether particular cell types are differentially abundant near *PIGR*⁺ or *PTGS1*⁺ tumour cells from patient 5. We believe this has substantially improved

the manuscript. We have added the following text and updated Figure 5 to highlight those populations with significant enrichment or depletion near tumour cells and subclones.

“Utilising significance testing by cell type label permutations, we consistently observed a significant enrichment of *CCL5+* T cells and *CXCL9+* macrophages in close proximity to both subclones across all distances. Conversely, there was a consistent and significant depletion of *LYZ+* macrophages and *SPP1+* macrophages near both subclones at all distances (**Fig. 5a, Supplementary Table 8**). We then conducted a comparison of heterotypic neighbours of the *PIGR+* and *PTGS1+* subclones, with significance determined through cell label permutation. The results indicated that across all three distances, *C1QC+* macrophages, *CCL5+* T cells, and *IGHG1+* plasma cells exhibited a significantly higher likelihood of association with the *PIGR+* clone. Conversely, *COL3A1+* fibroblasts showed a significantly higher likelihood of association with the *PTGS1+* clone (**Fig. 5b, Supplementary Table 8**).”

4. What was the number of transcripts/cell for the Cosmx data?

We have now added the following text to the methods section on CosMx

“Based on the cellpose segmentation results, we initially obtained 39,939 cells. After excluding cells with less than 100 transcripts, 26,895 cells were kept for downstream analysis. Across all FOVs 9,915,852 transcripts were detected; 80.97% of these were detected in cells with ≥ 100 transcripts. Cell segmentation doublets (artifactual segments containing merged signals from adjacent cells) were removed using Scrublet⁴⁴ which left 21,651 which were used for gene expression clustering, differential expression and cell-to-cell communication analyses.”

5. Can correlation analysis be done between spots and images in Cosmx vs Visium?

We believe in this case it would not be meaningful to do a correlation analysis between the Visium spots and the CosMx images; they are at different resolutions, they measure different numbers of genes, and they are not run on the same section (adjacent sections are used).

However, as part of the CosMx vs Visium comparison, **Figure 4d** shows the log fold changes estimated by CosMx and Visium between the *PIGR+* and *PTGS1+* subclones and the P5.1 and P5.2 malignant spots, respectively. We find that these values are indeed highly correlated between the two technologies, with Spearman’s correlation coefficient of 0.8 and p-value $< 2.2e-16$.

We have now added the correlation to the figure legend.

“The Visium and CosMx data were significantly correlated (Spearman’s correlation coefficient of 0.82 and p-value < 2.2e-16).”

Reviewers' Comments:

Reviewer #2:

Remarks to the Author:

The authors have clarified some of the questions, however two of my questions remain unanswered:

1. Why the pathologist-annotated tumor areas were not used in the clonal inference? They are difficult to see in Figure 1 – they should be represented in e.g. green colour for improved visualization. Further, if these were not used, the authors should at least justify why these annotations were not used?
2. My question about the sWGS analysis - Is the CNA profile difference between "clones" quantitatively significantly bigger than the inter-replicate heterogeneity? The estimated tumor fraction is also significantly lower than in P1.1., and it is suspicious that the predicted ploidy is around 2 in "normal" samples. Can the authors comment about this?

Reviewer #3:

Remarks to the Author:

The author have adequately addressed all of my comments.

Response to final review

Reviewer #2 (Remarks to the Author):

The authors have clarified some of the questions, however two of my questions remain unanswered:

1. Why the pathologist-annotated tumor areas were not used in the clonal inference? They are difficult to see in Figure 1 – they should be represented in e.g. green colour for improved visualization. Further, if these were not used, the authors should at least justify why these annotations were not used?

We attempted to use the pathologist annotations as suggested to annotate Visium spots with high or low tumour content and reran the copy number inference. To do this, we first mapped the location of tumour and non-tumour cells identified by Qupath to spots on the Visium slides (see below). We then ran inferCNV using expression from spots with QuPath tumour spot content of 0% as background and above 50% as tumour.

This resulted in substantially weaker and noisier copy number inference. Specifically there was evidence of CNA signal in the 'normal' spots and loss of signal in the 'tumour spots'.

To assess the impact of background sets we also ran inferCNV using the original background (RCTD tumour weight below 0.15) and spots with QuPath spot tumour percentage above 50%. This improves the inference, and predicts the same CNAs as in our original analyses, but reduces the number of spots. This suggests that the background spots annotated by QuPath as containing no tumour signal may actually overlap morphologically normal tumour cells. Delving into this further is beyond the scope of the current study.

As the reviewer's comments and our responses (including the above images), will be included upon publication we have not opted to incorporate a separate discussion on this into the main text.

In terms of recolouring the QuPath thumbnail images in **Figure 1e**. We agree they were difficult to visualise. We have removed the H&E background from the original QuPath images, made the tumour cells a bright red and kept the normal cells as pale green. We believe this has substantially improved the visualization. Note the original H&E is still included in **Fig. 1a**.

e QuPath tumour cells

In **Fig 2f** and **Supplementary figures S7f, S8d, S9f, S10f**, showing zoomed in regions, we keep the original QuPath images to make it clear which cells are being identified as tumour and normal.

2. My question about the sWGS analysis - Is the CNA profile difference between “clones” quantitatively significantly bigger than the inter-replicate heterogeneity?

Below are the Spearman correlations for all pairwise comparisons between the lpWGS informed ichorCNV CNA profiles. Also the tumour fraction and ploidy estimates from IchorCNA are shown.

	regions	Normal			P1.1		P1.2			P1.3		rank of replicate correlation
		1	2	3	1	2	1	2	3	1	2	
Normal	1		0.23	0.59	-0.04	-0.08	-0.1	-0.13	-0.18	-0.05	-0.04	1
	2	0.23		0.33	0.13	0.08	-0.06	-0.04	-0.03	0.01	0.1	1
	3	0.59	0.33		0.08	0.07	0.07	0.03	-0.06	0.03	0.1	1
P1.1	1	-0.04	0.13	0.08		0.81	0.77	0.82	0.8	0.82	0.82	4
	2	-0.08	0.08	0.07	0.81		0.74	0.76	0.68	0.68	0.66	1
P1.2	1	-0.1	-0.06	0.07	0.77	0.74		0.83	0.82	0.74	0.71	1
	2	-0.13	-0.04	0.03	0.82	0.76	0.83		0.89	0.74	0.75	1
	3	-0.18	-0.03	-0.06	0.8	0.68	0.82	0.89		0.73	0.74	1
P1.3	1	-0.05	0.01	0.03	0.82	0.68	0.74	0.74	0.73		0.82	1 *tie with P1.1 region 1
	2	-0.04	0.1	0.1	0.82	0.66	0.71	0.75	0.74	0.82		1 *tie with P1.1 region 1
IchorCNA estimates												
tumour fraction		0.181	0.246	0.203	0.592	0.908	0.535	0.635	0.714	0.679	0.69	
ploidy		2.22	1.98	1.95	2.05	1.97	2.01	1.73	2.18	2.1	2.25	

1. The CNA profiles from the tumour regions consistently correlate better with each other (green) than with the normal control regions (red).
2. The majority of the IchorCNA CNA profiles of replicates within each cluster were highly correlated. However for one P1.1 replicate (region 1), it correlated slightly better with replicates from the P1.3 and P1.2 clusters (Spearman 0.82), than its replicate (Spearman 0.81).

We have now added the following to the text to reflect this.

“Notably, the CNA profiles of replicates within each cluster were highly correlated with the exception of replicate 1 of P1.1 which had slightly stronger correlation with replicates from the P1.3 and P1.2 clusters (**Supplementary Table 3**).

Given the correlations above, the validation of the subclone specific CNAs in **Figure 2e**, and the prediction of subclones with mutually exclusive CNAs in both our data set and that of Ferri Borgogno we believe we have provided strong evidence of HGSOC subclones.

The estimated tumor fraction is also significantly lower than in P1.1., and it is suspicious that the predicted ploidy is around 2 in “normal” samples. Can the authors comment about this?

The estimated tumour fractions summarised above, show that P1.1 replicate 2 has the highest estimated tumour content (0.908) while the others range from 0.535 to 0.714. In contrast the normal samples were substantially lower, ranging from 0.181 to 0.246.

Regarding the ploidy estimates from IchorCNA, we don't think these are particularly informative. For normal a ploidy of 2 is expected.

Note IchorCNA actually requires setting an initial ploidy estimate parameter. Reviewing publications citing IchorCNA, the vast majority set the ploidy parameter to 2 (the default) and do not further mention or interpret the estimated ploidies. For our data we set --ploidy "c(2,3,4)" so initializations were tried at ploidies of 2, 3, and 4. The best solution from the IchorCNA model estimated values around 2 for both tumour and normal samples.

Reviewers' Comments:

Reviewer #2:

Remarks to the Author:

The authors have now addressed my comments. However, the new analyses performed (as requested already during the 1st round of revisions) highlight an important limitation of the study design potentially affecting the interpretation of the results. The authors now report, that the use of pathologist annotated tumor spots in the CNV inference actually failed to predict reliable tumor spatial genetic subclones.

This indicates that still further optimisation of the algorithm and CNV inference are needed especially regarding what spots are considered as the "tumor" and what as "normal".

In order to conform to the Journals transparency of reporting standards, it is critical to the readers to have the experimental design and potential limitations reported clearly in the manuscript. Therefore, I believe that after the below section in the manuscript, the authors should openly reported this somewhat contradictory result and address this limitation in the discussion.

"To confirm that our strategy reliably identified malignant clusters, we carried out a histopathological assessment of the H&E images from the Visium sections, which showed that in most cases tissue areas with low RCTD tumour cell scores corresponded to regions of cells labelled as stroma by a pathologist using Qupath, while high-confidence malignant CNA-based clusters corresponded to the regions of cells called as malignant (Fig. 1e)."

Response to fourth review

Reviewer #2 (Remarks to the Author):

The authors have now addressed my comments. However, the new analyses performed (as requested already during the 1st round of revisions) highlight an important limitation of the study design potentially affecting the interpretation of the results. The authors now report, that the use of pathologist annotated tumor spots in the CNV inference actually failed to predict reliable tumor spatial genetic subclones.

This indicates that still further optimisation of the algorithm and CNV inference are needed especially regarding what spots are considered as the "tumor" and what as "normal". In order to conform to the Journals transparency of reporting standards, it is critical to the readers to have the experimental design and potential limitations reported clearly in the manuscript. Therefore, I believe that after the below section in the manuscript, the authors should openly reported this somewhat contradictory result and address this limitation in the discussion.

"To confirm that our strategy reliably identified malignant clusters, we carried out a histopathological assessment of the H&E images from the Visium sections, which showed that in most cases tissue areas with low RCTD tumour cell scores corresponded to regions of cells labelled as stroma by a pathologist using Qupath, while high-confidence malignant CNA-based clusters corresponded to the regions of cells called as malignant (Fig. 1e)."

We have included the analysis and figures from the last response as a new **Supplementary note** and have added the following text where the reviewer suggested:

"Notably, repeating the inferCNV analyses using QuPath annotations to identify background spots overlapping morphologically normal cells resulted in substantially worse CNA inference and failed to predict subclones (**Supplementary Note 1**)."

In our opinion these results do not indicate a limitation of the approach with a need to optimise the algorithm. Instead they suggest that the transcriptome based decomposition with reference expression profiles approach is a more robust way to identify Visium spots sampling tumour cells than cellular morphology alone. In terms of transparency, the new supplementary note details the comparison and all methods used to generate our conclusions are detailed in the manuscript.